# When Diffusion Language Models Hesitate: Detecting and Correcting Visual Hallucinations via Confidence Fluctuation

Wenzheng Song [1]  Pei Chen [2]  Yichen Tan [1]  Zejian Li [3]  Lingyun Sun [2]

## Abstract

Multi-modal Diffusion Language Models (MDLMs) have emerged as a powerful alternative to autoregressive models in vision understanding, offering advantages in bidirectional context modeling and parallel decoding. However, existing MDLMs suffer from visual hallucinations due to the static nature of visual perception. Unlike autoregressive models, MDLMs lack the sequential dependency to dynamically interact with visual content. Therefore, MDLMs rely on fixed visual features encoded at initialization, causing the denoising process to drift toward language priors and lose its anchor to visual evidence. In this paper, we propose VGR (Visual-Guided Refinement), a framework that enables MDLMs to revisit visual details by exploiting diffusion dynamics. Our key insight is that the temporal trajectory of confidence during denoising reveals intrinsic uncertainty: while grounded tokens converge smoothly, hallucinated ones exhibit pronounced confidence fluctuation. VGR utilizes this fluctuation signal to detect uncertain spans and corrects them through targeted visual evidence extraction and in-place remasking. Extensive experiments on image captioning and hallucination evaluation benchmarks demonstrate that our method reduces hallucinations and recalls more details. We release our code at https://github.com/SongWZ3214/VGR.

## 1. Introduction

Multi-modal large language models (MLLMs) have achieved remarkable progress in vision-language under-

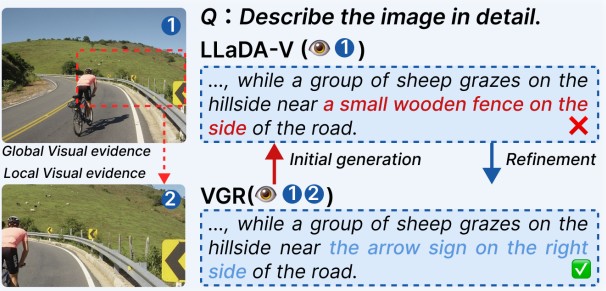

*Figure 1.* Comparison between existing MDLMs (e.g., LLaDA-V) and our proposed VGR framework. Traditional approaches rely solely on global image context for generation, often leading to hallucinations when describing fine-grained details. VGR first detects potential hallucinations and then performs targeted in-place rectification by consulting local visual evidence, ensuring the generated text is precisely grounded in visual details.

standing (Li et al., 2025c; Wang et al., 2025c). While existing MLLMs predominantly adopt autoregressive (AR) architectures for text generation, discrete diffusion models have recently emerged as an alternative paradigm (Li et al., 2025b). Due to bidirectional context modeling and parallel decoding, multi-modal diffusion language models (MDLMs) such as LLaDA-V (You et al., 2025), MMaDA (Yang et al., 2026), and LaViDa (Li et al., 2025a) offer distinct advantages over AR counterparts: high global consistency and generation efficiency (Li et al., 2025b; Nie et al., 2025; Ye et al., 2025).

Despite these architectural merits, MDLMs also face significant challenges of hallucinations (Guo & Tan, 2026). As Figure 1 shows, a model may confidently mistake the arrow sign for "a small wooden fence". We identify a fundamental cause rooted in the diffusion paradigm itself: the absence of continuous visual grounding. Existing MDLMs encode visual information into static features at initialization. In the subsequent denoising process, the model relies on this fixed visual representation without actively re-examining specific image regions, causing predictions to drift toward language priors (Favero et al., 2024; Chen et al., 2025) rather than remaining anchored to pixel-level evidence.

In autoregressive generation, token-level confidence has proven effective for uncertainty estimation, enabling low-

[1]College of Computer Science and Technology, Zhejiang University, Hangzhou, China [2]College of Artificial Intelligence, Zhejiang University, Hangzhou, China [3]School of Software Technology, Zhejiang University, Ningbo, China. Correspondence to: Pei Chen <chenpei@zju.edu.cn>.

*Proceedings of the 43rd International Conference on Machine Learning*, Seoul, South Korea. PMLR 306, 2026. Copyright 2026 by the author(s).

confidence tokens to be flagged for retrieval augmentation or self-correction (Zhang et al., 2025; Li et al., 2026a). More recent approaches dynamically re-examine visual regions through tool calls or interleaved visual tokens, thereby achieving "thinking with images" (Su et al., 2025; Zheng et al., 2026; Yu et al., 2025; Wang et al., 2025b). However, transferring these strategies to MDLMs faces structural barriers. The non-sequential generation dynamics preclude the assessment of reasoning progress, as intermediate tokens represent noisy predictions rather than meaningful partial outputs. In addition, the fixed sequence length prohibits dynamic insertion of visual tokens. Most critically, we find that final token confidence is unreliable in MDLMs: our pilot study reveals that hallucinated tokens frequently maintain high confidence scores, because the denoising process can converge to stable but incorrect predictions when language priors dominate over weak visual signals.

These limitations motivate us to seek uncertainty signals intrinsic to diffusion dynamics. Our key insight is that the temporal trajectory of confidence during denoising reveals information invisible in the final state. We observe that correctly grounded tokens exhibit smooth, monotonically increasing confidence, reflecting consistent convergence toward visual evidence. In contrast, hallucinated tokens display pronounced oscillations across denoising steps. Based on this analysis, we propose the *Confidence Fluctuation (CF)* metric to quantify trajectory instability. Guided by this, we propose **VGR** (**V**isual-**G**uided **R**efinement), a framework that enables MDLMs to revisit and rectify fine-grained visual details without architectural modifications. VGR leverages the intrinsic remasking capability of diffusion models through a three-stage pipeline: (1) **Uncertainty Localization**, which uses *CF* to pinpoint unstable text spans; (2) **Visual Evidence Extraction**, which utilizes an open-vocabulary detector to retrieve the localized regions; and (3) **Visual-Guided Refinement**, which remasks uncertain tokens and regenerates accurate descriptions based on visual evidence. In addition, we fine-tune the base model on a dataset constructed from RefCOCO+ (Kazemzadeh et al., 2014), ensuring that corrections are based on explicit visual evidence rather than language likelihood.

Our main contributions are summarized as follows:

- We identify confidence fluctuation as an intrinsic uncertainty signal for detecting fine-grained visual hallucinations in MDLMs.

- We introduce VGR, a framework that utilizes *CF* to guide targeted in-place rectification in MDLMs, leveraging dual-view visual evidence to resolve uncertain tokens without architectural modifications.

- Experiments across multiple MDLM backbones on benchmarks such as CapArena and AMBER demon-

strate that our method significantly reduces hallucinations and generalizes consistently across architectures.

## 2. Related Work

### 2.1. Multi-modal Diffusion Language Model

While autoregressive (AR) models currently dominate vision-language understanding with superior performance, discrete diffusion language models are emerging as a promising alternative (Li et al., 2025b; 2026b). Masked diffusion models such as LLaDA (Nie et al., 2025) and Dream7B (Ye et al., 2025) have demonstrated capabilities comparable to AR counterparts. These models conceptualize text generation as an iterative denoising process by predicting masked tokens. Multi-modal extensions including LLaDA-V (You et al., 2025), MMaDA (Yang et al., 2026), and LaViDA (Li et al., 2025a) further integrate vision encoders to project visual features into the language embedding space, realizing vision-language understanding and generation in diffusion paradigm. Benefiting from bidirectional attention and parallel decoding, multi-modal diffusion models (MDLMs) exhibit inherent architectural advantages in capturing global context and enhancing inference efficiency (Li et al., 2025b).

However, realizing this potential is hindered by severe hallucinations stemming from static visual perception. Since the model relies on a fixed visual representation encoded at initialization, it cannot actively re-examine visual details during the lengthy denoising trajectory, causing predictions to drift toward language priors. While diffusion models inherently support iterative refinement through remasking (Wang et al., 2025a), leveraging this capability for hallucination correction remains challenging. A recent attempt, ReDiff (Ji et al., 2025), trains LLaDA-V to self-correct errors via expert feedback, but relies on language-level signals and global images without explicitly re-grounding to fine-grained visual evidence. This motivates our investigation into reliable, visually-grounded uncertainty identification for targeted refinement.

### 2.2. Visual Evidence Guided Thinking

To enhance fine-grained visual perception and mitigate hallucinations, existing AR models have explored methods of introducing relevant visual evidence during generation. Early research provides global or local image descriptions to guide models in generating responses more faithful to the image (Li et al., 2025c; Xia et al., 2025). Chain-of-Thought (CoT) techniques further enable MLLMs to perceive images by multi-step reasoning (Zhang et al., 2024; Xu et al., 2025). However, these methods remain text-centric reasoning, lacking sufficient utilization and fine-grained perception of visual information (Su et al., 2025; Xu et al., 2026; Zheng et al., 2026).

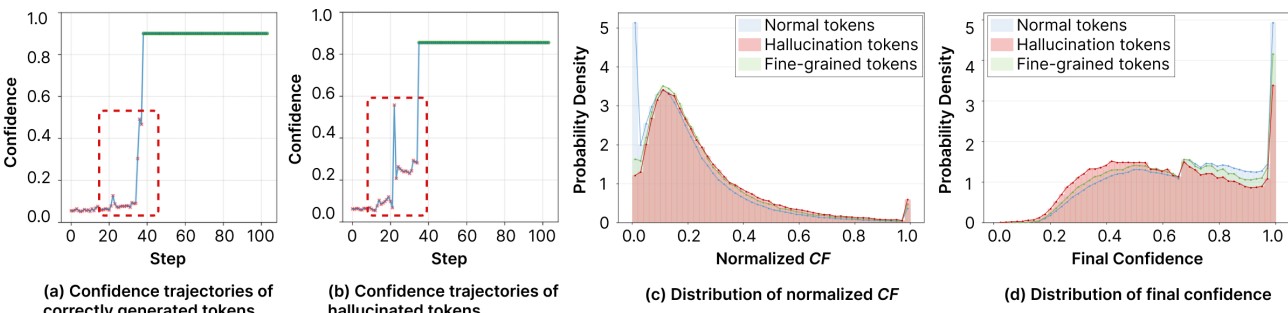

**(a) Confidence trajectories of correctly generated tokens.**

**(b) Confidence trajectories of hallucinated tokens.**

**(c) Distribution of normalized *CF***

**(d) Distribution of final confidence**

*Figure 2.* Analysis of confidence dynamics. (a) & (b) illustrate confidence trajectories: normal tokens exhibit a stable rise, while hallucinated tokens fluctuate significantly. (c) & (d) compare distributions: hallucinated tokens (red) clearly shift towards higher Confidence Fluctuation (*CF*) and lower final confidence.

Recent research enables models to "think with images", imitating human cognition that uses vision as a dynamic mental sketchpad. By constructing multi-modal interleaved CoT, this paradigm incorporates visual information into intermediate reasoning processes, thereby enhancing visual perception (Ni et al., 2025; Yu et al., 2025). In this way, visual information is transformed from a passive input into a dynamic, manipulable cognitive workspace (Su et al., 2025). In AR models, this is typically realized through tool-based region focusing (Zheng et al., 2026; Fan et al., 2026; Zeng et al., 2026), programmatic visual augmentation (Hu et al., 2024; Qiao et al., 2025), and intrinsic image generation (Wang et al., 2025b; Qin et al., 2026; Chern et al., 2025). However, transferring this paradigm to MDLMs presents structural challenges. Due to their non-autoregressive nature, MDLMs cannot spontaneously determine when visual evidence is needed. Furthermore, dynamically inserting visual tokens or generating tool calls remains difficult. Therefore, this work aims to investigate how to exploit the inherent remasking capability of MDLMs to identify when and which tokens should be corrected for visual hallucinations.

## 3. Preliminary

### 3.1. Multi-modal Diffusion Language Models

Unlike the token-by-token prediction of AR models, MDLMs formulate text generation through a forward and a reverse process. Formally, let $x_0 = [x_0^i]_{i=1}^N$ represent a clean sentence of $N$ tokens, with $I$ and $Q$ representing the input image and text prompt, respectively. In the forward process, a time step $t$ is uniformly sampled from $[0, 1]$. Each token in $x_0$ is replaced by a special mask token [MASK] with probability $t$, yielding a noisy sequence $x_t$. In the reverse process, MDLMs iteratively denoise from a fully masked sequence by predicting the original tokens. At each time step $t$, the model predicts probabilities $p_\theta(x_t|x_{t+1}, I, Q)$, updating all tokens in the entire sequence in parallel. We formally define the confidence $c_t^i$ of the $i$-th token as the maximum predicted probability assigned by the

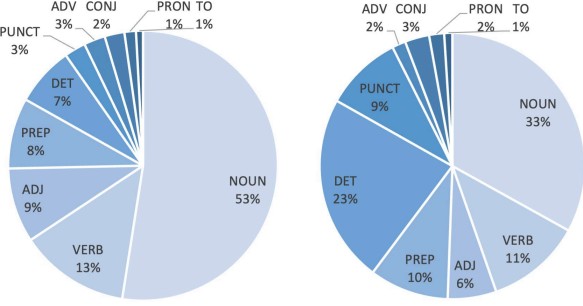

**(a) Part-of-Speech statistics of top-10 high CF tokens.**

**(b) Part-of-Speech statistics of top-10 low confidence tokens.**

*Figure 3.* Part-of-Speech (POS) Statistics. A comparison of the top-10 tokens with (a) Highest *CF* and (b) Lowest Final Confidence. High-*CF* tokens are dominated by semantic classes (Nouns, Verbs) relevant to visual content, while Low-Confidence tokens contain significantly more syntactic functional classes (Determiners, Punctuation).

network:

$$c_t^i = \max_{v \in \mathcal{V}} p_\theta(x_t^i = v|x_{t+1}, I, Q), \qquad (1)$$

where $\mathcal{V}$ denotes the vocabulary. Based on this metric, high-confidence tokens are unmasked, while the remaining tokens are remasked and predicted in subsequent steps. While this diffusion architecture offers the advantage of efficient parallel generation, it hinders the model's ability to dynamically focus on new visual regions during the generation process. Fortunately, the remasking capability of diffusion models provides a natural mechanism for correction: uncertain tokens can be remasked and regenerated with additional visual guidance. The key challenge lies in reliably identifying which tokens require such intervention.

### 3.2. Empirical Analysis

**Limitations of Static Confidence.** Token confidence is widely used as an uncertainty indicator in AR models (Li et al., 2025c). Motivated by this, we initially hypothesized

that visual hallucinations in MDLMs would similarly exhibit low final confidence. To test this, we conducted a pilot study on DetailCaps-4870 (Dong et al., 2024) using LLaDA-V (You et al., 2025). We employed Gemini-2.5-flash (Comanici et al., 2025) to annotate generated captions under strict criteria: (1) **Hallucinated Tokens** are the smallest semantically complete phrases that factually contradict the image; (2) **Fine-grained Visual Tokens** describe specific attributes (e.g., color, texture) or complex spatial relations, distinguishing them from generic object labels.

Contrary to our hypothesis, the results revealed no significant overlap between tokens with low final confidence and annotated hallucinations. We argue that in the non-sequential diffusion process, high final confidence merely indicates convergence at the final step, masking the model's earlier uncertainty during the denoising trajectory.

**Theoretical Analysis: Dynamic Evidence Competition.** To capture this hesitation, we analyze the generation dynamics from the perspective of posterior estimation evolution. In discrete diffusion, the predicted logits $\mathbf{z}_t$ for a token can be modeled as a time-varying weighted combination of two information sources:

$$\mathbf{z}_t \approx \alpha(t) \cdot \underbrace{\phi_{\text{lang}}(x_t, Q)}_{\text{Language Prior}} + \beta(t) \cdot \underbrace{\phi_{\text{vis}}(x_t, I)}_{\text{Visual Evidence}}, \qquad (2)$$

where $\phi_{\text{lang}}$ captures linguistic plausibility, and $\phi_{\text{vis}}$ represents visual grounding. Crucially, the weights $\alpha(t)$ and $\beta(t)$ evolve dynamically. In the early high-noise stages, visual attention lacks context, causing the model to default to language priors ($\alpha(t) \gg \beta(t)$). As denoising progresses, visual evidence gains dominance ($\beta(t) \uparrow$), forcing a shift toward pixel-level reality. In hallucination scenarios, these sources are antagonistic: the probability distribution must migrate from the prior-driven mode to the evidence-driven mode on the probability simplex. This transition causes a transient dip in confidence before reconvergence.

This competition manifests as non-monotonic updates in the probability trajectory. To capture this, we introduce Confidence Fluctuation (*CF*) to quantify the accumulated deviation of the confidence trajectory:

$$CF^i = \sum_{t=1}^{T} |c_{t-1}^i - c_t^i| - (c_0^i - c_T^i), \qquad (3)$$

where $c_t^i$ is the confidence of token $i$ at step $t$, following the convention in Section 3.1 that $t = 0$ denotes the final clean prediction and $t = T$ the fully masked initialization. The first term accumulates the total variation of the confidence trajectory over all adjacent steps, while the second term $(c_0^i - c_T^i)$ subtracts the net confidence gain from initialization to the clean state. A high *CF* thus reflects non-monotonic oscillation that cannot be explained by smooth convergence.

We computed normalized *CF* scores (via min-max scaling within each sequence, see Appendix A) and compared their distribution against final confidence, as shown in Figure 2 (c) and (d). Key findings include: (1) The results reveal that hallucinated tokens exhibit a distinct shift toward high *CF*, manifesting as a heavier tail distribution where the probability density of hallucinations significantly exceeds that of normal tokens. (2) Crucially, statistical analysis of the top-10 ranked tokens demonstrates a mere 15.73% overlap between the high-*CF* set and the low-confidence set. (3) A Part-of-Speech analysis (Figure 3) further clarifies this divergence: the high-*CF* group is dominated by content words (nouns/verbs) linked to visual semantics, while the low-confidence group is skewed toward functional tokens. To further verify the reliability of *CF*, we evaluate the base model under varied denoising steps $T \in \{32, 64, 128\}$, and find that these findings hold consistently across settings. The detailed results can be seen in Appendix A. This confirms that while low confidence reflects syntactic uncertainty, *CF* serves as a reliable and specialized indicator for visual epistemic uncertainty arising from multi-modal conflicts.

## 4. Visual-Guided Refinement

Based on the observation that *CF* serves as a reliable proxy for visual epistemic uncertainty, we propose the Visual-Guided Refinement (VGR) framework to resolve such "hesitations" by re-integrating fine-grained visual evidence into the denoising process. Our approach comprises a training phase to empower the MDLM with dual-view perception capabilities and an inference-time refinement loop that automatically localizes, remasks, and corrects uncertain tokens guided by targeted local details.

### 4.1. Dual-View Perception Training

To empower the model with the ability to handle dual-view inputs and rectify text via visual evidence, we construct the VGR-Instruct dataset and fine-tune an MDLM.

**Data Construction Pipeline.** Current multi-modal datasets are predominantly limited to global descriptions, suffering from a lack of "local-global" correspondence. We bridge this gap by building a dataset comprising about 37k samples derived from RefCOCO+ (Kazemzadeh et al., 2014). We select RefCOCO+ as the source because its expressions focus on object attributes (e.g., color, shape) rather than absolute location descriptions, which aligns with our goal of fine-grained visual perception. The construction process involves three steps: (1) **Instance Sampling:** For each image, we sample up to two object instances. Each instance corresponds to a bounding box (bbox) and its longest referring expression from the annotation list. (2) **Caption Synthesis:** We utilize an advanced MLLM (e.g., Gemini-2.5-flash (Comanici et al., 2025)) to generate a global description for

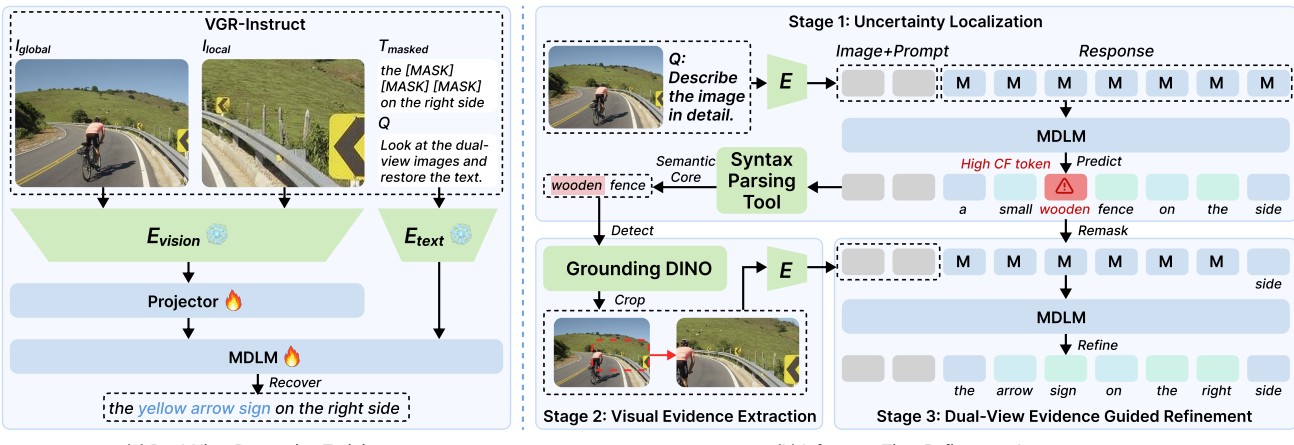

*Figure 4.* The overall VGR framework. (a) Training: We construct the VGR-Instruct dataset to fine-tune the MDLM with dual-view inputs (global image and local crop) to recover masked local descriptions. (b) Inference: The refinement pipeline consists of three stages: (1) Uncertainty Localization detects high-*CF* tokens and anchors them to semantic cores using syntax parsing; (2) Visual Evidence Extraction employs Grounding DINO to locate and crop the corresponding image region with visual gating; (3) Dual-View Guided Refinement remasks the uncertain span and regenerates it conditioned on both global and local visual evidence.

each sample, which is prompted to integrate the detailed attributes of the specified objects. (3) **Masking and Cropping:** We employ a matching algorithm to locate the object descriptions $T_{local}$ within the global caption $T_{global}$ and replace them with mask tokens, yielding $T_{masked}$. The detailed algorithm can be found in Appendix B. Simultaneously, we crop the corresponding regions $I_{local}$ from the original image $I_{global}$ based on the bboxes to serve as visual evidence. The final data instance is formatted as a quintuple: $(I_{global}, I_{local}, T_{global}, T_{masked}, T_{local})$.

**Training Strategy.** We use the constructed dataset to fine-tune the model, teaching it to recover $T_{local}$ from $T_{masked}$ conditioned on $I_{global}$ and $I_{local}$. To preserve the model's fundamental capabilities, we mix in a small amount of global caption generation data. The training objective can be modeled as a unified task of minimizing the reconstruction loss in masked regions:

$$\mathcal{L}(\theta) = -\mathbb{E}_{\mathcal{D}} \left[ \frac{1}{|\mathbf{M}|} \sum_{i \in \mathbf{M}} \log p_\theta(x_0^i \mid x_M, \mathcal{I}) \right], \quad (4)$$

where $x_M$ is the masked sequence, $x_0$ is the clean sequence, $\mathbf{M}$ is the special mask token, and $\mathcal{D} = \{\mathcal{I}, x_0, x_M\}$. $\mathcal{I}$ is the multi-modal context which can be formulated as:

$$\mathcal{I} = \begin{cases} \{I_{global}, Q\}, & for\ captioning, \\ \{I_{global}, I_{local}, Q\}, & for\ refinement, \end{cases} \quad (5)$$

where $Q$ is the text prompt. This mixed training method encourages the model to utilize local visual features for text rectification while maintaining global semantic coherence.

## 4.2. Inference-Time Refinement Loop

**Stage 1: Uncertainty Localization.** We employ *CF* to measure visual uncertainty in generation. However, simple thresholding may extract trivial tokens like prepositions, failing to capture the semantic core. To precisely pinpoint the visual entities requiring correction, we propose a syntax-aware localization strategy:

- **Fluctuation Screening:** First, the MDLM generates an initial text. We calculate the *CF* scores for all tokens and identify the token $x_t^{cf}$ with the highest fluctuation as the primary candidate.

- **Semantic Anchoring:** Using a syntax parsing tool like spaCy, we map $x_t^{cf}$ to its corresponding visual entity. We first check if $x_t^{cf}$ resides within a pre-defined noun chunk. If so, the entire chunk is immediately locked as the semantic anchor $A$. If $x_t^{cf}$ is isolated, we traverse the dependency tree based on POS tags. For verbs, we identify the subject or object associated with the action; for prepositions, we locate the object; for modifiers like adjectives, we perform a recursive head traversal to handle nested attributes, tracing the dependency chain upward until a nominal ancestor is reached.

- **Union Masking:** To ensure grammatical coherence and sufficient context, we define the final mask as the spatial union of the fluctuation token and the semantic anchor. The mask range is determined by $[\min(S(x_t^{cf}), S(A)), \max(E(x_t^{cf}), E(A))]$, where $S$ is the start and $E$ is the end.

**Stage 2: Visual Evidence Extraction.** We utilize an open-vocabulary detector, Grounding DINO (Liu et al., 2024b),

*Table 1.* The main results on three fine-grained image caption benchmarks and two hallucination evaluation benchmarks. The best scores of MDLMs are in **bold**. Results in the first three columns (marked with †) for all baseline models are taken from (Ji et al., 2025).

| Model | CapMAS † | CapArena † | DetailCaps-4870 † | AMBER-g | | | | MMHal-Bench | |
|---|---|---|---|---|---|---|---|---|---|
| | Avg. Score ↑ | CapArena-Auto ↑ | CAPTURE ↑ | CHAIR ↓ | Cover ↑ | Hal ↓ | Cog ↓ | Score ↑ | Hal-Rate ↓ |
| *AR model* | | | | | | | | | |
| LLaVA-1.5-7B (Liu et al., 2024a) | 49.73 | -94.00 | 51.08 | 7.8 | 51.0 | 36.4 | 4.2 | 1.82 | 0.66 |
| InternVL-2.5-7B (Chen et al., 2024) | 69.88 | -29.83 | 57.80 | 5.2 | 54.3 | 32.9 | 1.8 | 2.99 | 0.47 |
| Qwen2.5-VL-7B (Bai et al., 2025) | 73.51 | -16.83 | 60.61 | 10.3 | 65.9 | 48.6 | 3.2 | 2.26 | 0.73 |
| *Discrete diffusion model* | | | | | | | | | |
| MMaDA (Yang et al., 2026) | 35.92 | -97.00 | 19.55 | 44.9 | 26.0 | 72.8 | 6.4 | 0.91 | 0.78 |
| FUDOKI (Wang et al., 2026) | 45.72 | -98.83 | 57.92 | 9.8 | 51.9 | 41.6 | 3.4 | 1.38 | **0.60** |
| LaViDa (Li et al., 2025a) | 51.32 | -90.00 | 57.28 | 8.0 | 51.4 | 40.2 | 3.5 | 1.58 | 0.61 |
| LLaDA-V (You et al., 2025) | 58.61 | -77.17 | 59.62 | 8.2 | 61.8 | 44.9 | 4.2 | 1.98 | 0.75 |
| LLaDA-VGR | **60.84** | **-52.17** | **62.35** | **6.0** | **64.5** | **37.2** | **2.2** | **2.92** | **0.60** |

to map the uncertain token span to a bounding box within the image. To prevent the model from generating new hallucinations due to unclear local visual evidence, we introduce a visual gating mechanism. The subsequent refinement is executed only when the detector's confidence surpasses a preset threshold $\tau$. Otherwise, the local visual evidence is deemed insufficient, and the refinement for that round is only guided by the global image. For visual regions that pass the gate, we crop and proportionally expand them to obtain a local view $I_{local}$. Specifically, we apply a $1.5\times$ expansion scale to regions occupying less than $5\%$ of the image area to preserve essential environmental context.

**Stage 3: Dual-View Evidence Guided Refinement.** In this stage, the span with the highest *CF* is remasked along with $k$ adjacent context tokens. This expansion creates a semantic buffer, allowing the MDLM to regenerate the region with greater flexibility and coherence. To address the issue of context loss caused by relying solely on local images, we adopt a dual-view input strategy. Specifically, we input both $I_{global}$ and $I_{local}$ into the MDLM, employing a shared vision encoder and projector to extract features. We adopt sequence concatenation, where local vision tokens are appended to global vision tokens. This enables the MDLM to attend to fine-grained details via its native bidirectional attention, while preserving the original architecture. To ensure accurate alignment between the local visual evidence and the refinement target, we process only one span per round. After the refinement, the *CF* values of tokens and candidate spans are updated for the next turn. We set a maximum number of refinement turns to prevent over-editing and trade off computational efficiency. The refinement loop terminates when the maximum number of iterations is reached.

## 5. Experiments

### 5.1. Experiment Settings

**Benchmarks.** We evaluate on five datasets across two categories to comprehensively assess fine-grained perception

and hallucination mitigation: (1) *Fine-grained Image Captioning.* CapMAS (Lee et al., 2025) uses *Coverage* and *Factuality* to evaluate the trade-off between information density and accuracy, while using *CLAIR* to evaluate the overall caption quality. CapArena (Cheng et al., 2025) is a pairwise preference benchmark that compares the response of the test model and baseline models. It uses the *CapArena-Auto* to measure the win ratio, which reflects the caption quality. DetailCaps-4870 (Dong et al., 2024) provides high-quality detailed references, utilizing the *CAPTURE* metric to evaluate the recall of visual elements such as objects, attributes, and relations. (2) *Hallucination Evaluation.* AMBER-g (the generative task subset of AMBER (Wang et al., 2024)) uses four metrics: *CHAIR* measures the frequency of hallucinatory objects appearing in the responses; *Cover* measures the object coverage of responses; *Hal* represents the proportion of responses with hallucinations; *Cog* assesses whether the hallucinations in MLLMs are similar to those in human cognition. MMHal-Bench (Sun et al., 2024) evaluates hallucinations in open-ended responses, complementing existing benchmarks with visual question answering (VQA) task evaluations. It reports an average quality *score* (0-6) and hallucination rate (*Hal-Rate*) to quantify the factuality of the generated content relative to the visual context.

**Baselines.** We compare against four state-of-the-art MDLMs: LLaDA-V (You et al., 2025), MMaDA (Yang et al., 2026), LaViDa (Li et al., 2025a), and FUDOKI (Wang et al., 2026). We also select representative AR models including LLaVA-1.5-7B (Liu et al., 2024a), InternVL-2.5-7B (Chen et al., 2024) and Qwen2.5-VL-7B (Bai et al., 2025) for comparison.

**Implementation Details.** We build upon LLaDA-V (You et al., 2025) with a mixed tuning strategy: the vision projector is fully fine-tuned while the language backbone uses LoRA (Hu et al., 2022) on attention and FFN modules. Training and inference are both performed on a single node with four 80GB A100 GPUs. Detailed hyper-parameters are provided in Table 9 in Appendix C.

*Table 2.* Ablation study of component contributions.

| Model Version | CapArena | AMBER-g | | | |
|---|---|---|---|---|---|
| | CapArena-Auto ↑ | CHAIR ↓ | Cover ↑ | Hal ↓ | Cog ↓ |
| LLaDA-V | -77.17 | 8.2 | 61.8 | 44.9 | 4.2 |
| LLaDA-V (SFT) | -61.33 | 6.2 | 63.7 | 38.5 | 2.4 |
| LLaDA-VGR | **-52.17** | **6.0** | **64.5** | **37.2** | **2.2** |

*Table 3.* Comparison of uncertainty localization strategies, visual evidence granularity, and remasking strategies. The default settings, which serve as control variables across other ablation groups, are highlighted in **bold**.

| Strategy | CapArena | AMBER-g | | | |
|---|---|---|---|---|---|
| | CapArena-Auto ↑ | CHAIR ↓ | Cover ↑ | Hal ↓ | Cog ↓ |
| Low Confidence | -57.83 | **6.0** | 64.5 | 36.9 | 2.4 |
| **High *CF*** | **-52.17** | **6.0** | 64.5 | 37.2 | **2.2** |
| Hybrid | -56.50 | **6.0** | 64.9 | 36.0 | **2.2** |
| Global only | -53.67 | 6.1 | 64.4 | **37.2** | **2.2** |
| Local only | -56.67 | 6.3 | 64.1 | 38.4 | 2.4 |
| **Dual-View** | **-52.17** | **6.0** | **64.5** | **37.2** | **2.2** |
| Single Token | -56.00 | 6.2 | 64.1 | 37.1 | 2.4 |
| Fixed Window | -59.50 | 6.2 | **64.6** | **37.0** | 2.3 |
| **Syntax-aware** | **-52.17** | **6.0** | 64.5 | 37.2 | **2.2** |

## 5.2. Main Results

Table 1 presents results across all five benchmarks. LLaDA-VGR achieves state-of-the-art performance among MDLMs and substantially narrows the gap with leading AR models.

On the CapMAS benchmark, LLaDA-VGR achieves an average score of 60.84, surpassing the baseline LLaDA-V by 2.23 points. The detailed scores on three metrics can be found in Table 10 in Appendix D. On CapArena, our LLaDA-VGR shows a remarkable 25.00 point improvement over LLaDA-V, indicating that the generated captions are preferred for their richness and accuracy. On the DetailCaps-4870 benchmark, LLaDA-VGR achieves a CAPTURE score of 62.35, surpassing the baseline LLaDA-V by 2.73 points. This demonstrates our model's superior capability in recalling fine-grained visual elements.

LLaDA-VGR also demonstrates remarkable robustness in reducing hallucinations. On the generative tasks in AMBER benchmark, the three hallucination metrics, CHAIR, Hal, and Cog, decrease by 2.2, 7.7, and 2.0 compared with LLaDA-V, respectively. At the same time, the Cover metric increases from 61.8 to 64.5, indicating the more detailed perception ability of LLaDA-VGR. Similarly, on MMHal-Bench, we achieve the lowest Hal-Rate of 0.60 tied with FUDOKI and the highest Score of 2.92 among MDLMs. These results validate that our work effectively rectifies generative hallucinations in MDLMs by grounding visual evidence.

Notably, even compared to strong AR baselines like Qwen2.5-VL-7B, our method exhibits competitive perfor-

mance on DetailCaps-4870, AMBER and MMHal-Bench, highlighting the potential of diffusion models in reliable visual reasoning.

## 5.3. Ablation Studies

**Effectiveness of Components.** Table 2 verifies the contributions of our training and inference strategies. We integrate the fine-tuned model with the standard denoising pipeline in LLaDA-V (without inference-time refinement loop), obtaining LLaDA-V (SFT). The results demonstrate that fine-tuning on VGR-Instruct significantly improves the model's performance. For example, CapArena-Auto score increases by 15.84 and Cog decreases from 4.2 to 2.4 after fine-tuning, validating the efficacy of learning explicit global-local visual alignment. Integrating the VGR inference loop further improves the performance on image understanding and hallucination mitigation. Notably, while this step boosts CapArena-Auto to -52.17 and suppresses Hal to 37.2, it concurrently improves Cover from 63.7 to 64.5. This demonstrates that our refinement actively recovers missed details rather than merely trading recall for safety.

**Uncertainty Localization Strategy.** In the first block of Table 3, we compare different metrics for identifying uncertain tokens. Overall, our high *CF* strategy proves superior, significantly outperforming the standard low confidence approach in detailed caption quality while achieving comparable hallucination mitigation. To investigate whether these two signals could complement each other, we evaluate a hybrid strategy. This method is implemented sequentially by dedicating the first half of the refinement turns to high *CF* candidates, followed by low confidence candidates for the remaining half, within the same total iteration budget. The results present a trade-off: the hybrid strategy achieves the best performance on the AMBER-g benchmark, indicating that combining signals effectively expands the recall of potential errors. However, the hybrid strategy leads to a notable drop on CapArena. This suggests that Low Confidence is a noisy indicator: it frequently flags trivial tokens and rare but correct entities, leading to unnecessary or detrimental refinements that disrupt linguistic fluency. Consequently, *CF* provides a safe uncertainty localization metric, which offers precise hallucination mitigation without compromising the naturalness of generation.

**Visual Evidence Granularity.** In the second block of Table 3, we analyze the impact of different visual evidence granularity in the refinement stage. The results demonstrate that only providing local visual evidence suffers from a lack of global context, leading to suboptimal performance. Our dual-view strategy achieves the best trade-off, surpassing the global-only strategy on CapArena while maintaining less hallucination. To intuitively visualize the specific contribution of dual-view evidence to fine-grained perception, we

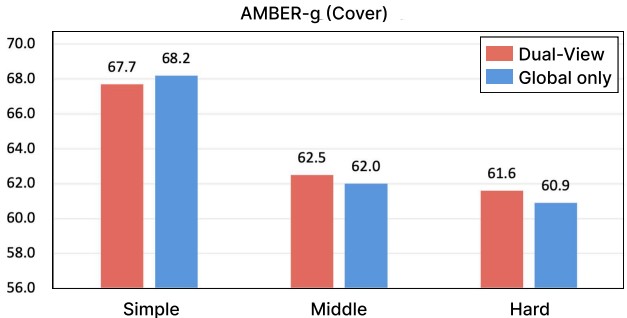

*Figure 5.* Performance comparison of AMBER-g Cover metric across difficulty levels. The test set is stratified by object scale. While the Global-only baseline suffices for large objects (Simple), Dual-View outperforms it on the Hard subset (small objects), demonstrating its superior capability in perceiving and recovering fine-grained details that are overlooked in the global view.

stratified the AMBER dataset into simple, middle, and hard subsets based on object scale and plotted the Cover scores in Figure 5 (complete metrics are provided in Table 13 in Appendix D). The trend in Figure 5 reveals a critical insight regarding the mechanism's utility: on the simple subset, the global-only strategy performs slightly better, likely because the global resolution is sufficient and avoids potential multi-view alignment noise. However, as the object scale decreases, the advantage shifts significantly. On the Hard subset, the dual-view strategy demonstrates a clear gain in coverage. This confirms that our mechanism acts as a "magnifying glass", effectively recovering fine-grained details for small objects that are otherwise blurred in the global view, thereby enhancing the model's recall capabilities in complex scenarios without compromising general performance.

**Remasking Strategy.** In the last block of Table 3, we compare different remasking strategies. Remasking a single token yields suboptimal performance on hallucination metrics. This is likely because isolated tokens lack complete semantic context, making it difficult to retrieve effective visual evidence. Compared to the fixed window strategy, our syntax-aware approach maintains comparable and even superior hallucination mitigation, while significantly boosting the performance on CapArena. By anchoring uncertain tokens to their semantic cores, our method preserves syntactic coherence during refinement, leading to more accurate corrections. In addition, we investigate the influence of the remasking window size $k$ in Table 14 in Appendix D and observe that $k = 2$ yields the optimal performance.

**Visual Gate Threshold.** We further evaluate the impact of the visual gate threshold $\tau$ that filters low-quality detection results on dataset AMBER. Beyond the standard *Cover* metric, we introduce two complementary metrics to capture the practical behavior of the gate: (1) **Correction Rate**, defined as the proportion of hallucinations that are

*Table 4.* Ablation on the visual gate threshold $\tau$ on AMBER. Higher Cover and Correction Rate indicate stronger fine-grained recovery, while a lower Over-correction Rate indicates safer refinement. The best scores are highlighted in **bold**.

| $\tau$ | Cover ↑ | Correction Rate ↑ | Over-correction Rate ↓ |
|---|---|---|---|
| 0.2 | **65.2** | 97.2% | 0.536% |
| 0.4 | 65.1 | **97.4%** | **0.521%** |
| 0.6 | 64.9 | 93.3% | 0.532% |

*Table 5.* Generalization of VGR to LaViDa and MMaDA. VGR improves both base models across captioning and hallucination metrics, though the magnitude of improvement differs across architectures.

| Model | CapArena | | AMBER-g | | |
|---|---|---|---|---|---|
| | CapArena-Auto ↑ | CHAIR ↓ | Cover ↑ | Hal ↓ | Cog ↓ |
| LaViDa | -90.00 | 8.0 | 51.4 | **40.2** | 3.5 |
| LaViDa-VGR | **-89.50** | **7.9** | **60.0** | 45.5 | **2.9** |
| MMaDA | -97.00 | 44.9 | 26.0 | 72.8 | 6.4 |
| MMaDA-VGR | **-96.00** | **10.5** | **54.9** | **39.6** | **2.6** |

successfully rectified after refinement, which reflects the recall of effective corrections; (2) **Over-correction Rate**, defined as the proportion of originally correct tokens that are mistakenly modified during refinement, which reflects the risk of corrupting grounded content under noisy detection. Table 4 reports results across $\tau \in \{0.2, 0.4, 0.6\}$. Lower thresholds increase over-correction risk due to detector noise. Conversely, higher thresholds cause the model to skip subtle details, reducing the correction rate. The value of 0.4 provides the balance.

### 5.4. Generalization to Other MDLMs

To verify that VGR is not tied to a specific MDLM architecture, we further apply both the dual-view perception training and the inference-time refinement loop to two additional masked diffusion models, LaViDa (Li et al., 2025a) and MMaDA (Yang et al., 2026). The training data, hyper-parameters, and inference configurations are kept consistent with our LLaDA-V setup; full implementation details are provided in Appendix C. As shown in Table 5, VGR brings meaningful improvements to both base models. MMaDA, whose unified discrete tokenization yields the weakest fine-grained perception, benefits most across all five metrics. LaViDa improves on captioning quality and coverage, but exhibits a Hal increase. This might be a recall–precision trade-off: as VGR drives the model to recover more fine-grained details, the denser descriptions expose more opportunities for error, which a base model with weaker fine-grained perception is less able to ground reliably. These results confirm that VGR is broadly applicable across masked MDLMs, and further suggest that its effectiveness might scale with the base model's capacity for reliable visual re-grounding.

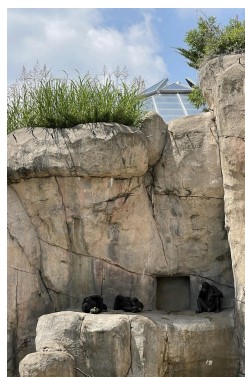

**Initial generation**
The image captures a serene outdoor scene under a clear blue sky, featuring a prominent rock formation with a large, textured cliff face. On the left, a cluster of green plants is nestled among the rocks, adding a touch of greenery to the scene. To the right, a building with a glass roof is partially visible, suggesting the presence of a structure behind the cliff. Two black objects, which appear to be bags or backpacks, are resting side by side on the rocks near the edge of the cliff. In the foreground, a few smaller rocks are visible at the base of the cliff, adding to the natural ambiance of the setting.

**Round 1**
→ High CF token: "backpacks"
→ Semantic core: "backpacks"
→ Applied masking on: "bags or backpacks"
→ Refinement result: "bags or containers"

**Round 2**
→ High CF token: "resting"
→ Semantic core: "Two black objects"
→ Applied masking on: "Two black objects, which appear to be bags or containers, are resting side by"
→ Refinement result: "Three dark-colored animals, possibly gorillas, are seen laying down on their"

...

**Final result**
The image captures a serene outdoor scene under a clear blue sky, featuring a prominent rock formation with a large, textured cliff face. On the left, a cluster of green plants is nestled among the rocks, adding a touch of greenery to the scene. To the right, a building with a glass roof is partially visible, suggesting the presence of a structure or facility nearby. Three dark-colored animals, possibly gorillas, are seen laying down on a patch of flat rocks near the edge of the cliff. In the foreground, a few smaller rocks are visible at the base of the cliff, adding to the natural ambiance of the setting.

*Figure 6.* Qualitative visualization of the iterative refinement process. VGR employs high *CF* to pinpoint hallucinations (red), retrieves local visual evidence, and iteratively rectifies the description into a grounded output (blue).

## 5.5. Qualitative Analysis

**Visual Refinement Process.** Figure 6 illustrates a challenging case where VGR corrects a semantic hallucination through iterative refinement. Initially, the model misidentifies dark shapes on the rocks as "bags or backpacks", likely due to their static posture and lack of fine-grained features in the global view. In the first round, the token "backpacks" triggers a high *CF* alert. While the initial refinement to "containers" remains imperfect, the persistent confidence fluctuation drives a second round of correction. By re-examining the local visual evidence with an expanded semantic core ("two black objects"), the model successfully recognizes the objects as "Three dark-colored animals, possibly gorillas" and corrects the count.

**Advantage over Low Confidence.** This case highlights a critical advantage of our method. As detailed in Figure 8 in Appendix E, the low confidence strategy initially targets irrelevant tokens (e.g., "building"). It fails to prioritize the hallucinated "backpacks" because the phrase is linguistically plausible and thus assigned high confidence. In contrast, our *CF* metric captures the visual ambiguity hidden behind high language priors, allowing the model to question and correct confident hallucinations.

## 5.6. Inference Cost Analysis

We conduct a comprehensive inference cost analysis. In terms of latency, LLaDA-VGR introduces a $2.1\times$ time overhead, with each refinement round costing only 3.30s, as the model only needs to denoise specific masked spans rather than regenerating the full sequence. The pipeline also supports batch inference, which further amortizes the overhead. We test on MMHal-Bench and the per-sample latency reduces from 24.06s to 7.67s ($3.14\times$ speedup) when scaling the batch size from 1 to 8. In terms of memory, the peak GPU memory usage at batch size 1 is 28.29 GB for LLaDA-VGR versus 23.50 GB for the base LLaDA-V, remaining

well within standard hardware budgets. Detailed results are provided in Appendix D.

## 5.7. Limitations and Future Work

VGR is designed to detect and correct visual hallucinations in a way that aligns with the architectural characteristics of MDLMs, achieved through the integration of an external visual detector and an iterative refinement loop. Therefore, the accuracy of hallucination correction partially depends on the reliability of external tools such as spaCy and Grounding DINO. Although visual gating and proportional crop expansion mitigate error propagation, the quality of refinement is ultimately bounded by the perception capability of these auxiliary modules. In addition, VGR operates as an inference-time augmentation rather than an architectural solution. The underlying tendency of MDLMs to drift toward language priors is alleviated at generation time but not eliminated at the model level. A promising direction is to internalize visual re-grounding as an intrinsic behavior of the denoising process, eliminating the reliance on an external refinement loop. We leave this for future work.

## 6. Conclusion

In this paper, we present VGR, a framework enhancing fine-grained visual grounding in MDLMs. To address the unreliability of static confidence, we introduce confidence fluctuation to localize hallucinations by capturing temporal instability during denoising. We first empower the MDLM with Dual-View Perception through fine-tuning on our curated VGR-Instruct dataset. At inference time, the model autonomously localizes uncertainties via syntax-aware anchoring, retrieves local visual evidence, and executes targeted in-place correction. Experimental results demonstrate that VGR achieves state-of-the-art performance among MDLMs, significantly mitigating hallucinations in fine-grained perception tasks. This work advances the development of robust, visually-faithful diffusion-based multi-modal systems.

## Acknowledgements

This research was supported by the National Natural Science Foundation of China (No.62502436).

## Impact Statement

This paper aims to enhance the reliability and faithfulness of Multi-modal Diffusion Language Models (MDLMs) by mitigating visual hallucinations and improving fine-grained visual grounding. By aligning generated descriptions more closely with visual evidence, our method contributes to the development of more trustworthy multi-modal AI systems, with potential benefits for downstream applications such as image-based information access and content verification. Like other research on generative models, our work shares broader concerns regarding the use of automated content generation. However, since VGR is designed to ground outputs in visual evidence rather than generate novel content, it tends to reduce, rather than amplify, the risk of unfaithful generation.

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

# A. Empirical Study Details

**Calculation of Normalized Confidence Fluctuation.** While Equation (3) in the main text defines the raw Confidence Fluctuation (*CF*) for a single token, the absolute magnitude of these values can vary significantly across different samples due to varying sequence lengths and image complexities. To mitigate the influence to the distribution statistics brought by this difference, we apply min-max normalization to the raw *CF* scores within each generated sequence.

Formally, let $\mathbf{CF} = \{CF^1, CF^2, \ldots, CF^N\}$ denote the set of raw *CF* scores for a generated sequence of length $N$. The normalized score $\hat{CF}^i$ for the $i$-th token is calculated as:

$$\hat{CF}^i = \frac{CF^i - \min(\mathbf{CF})}{\max(\mathbf{CF}) - \min(\mathbf{CF})} \tag{6}$$

where $\min(\mathbf{CF})$ and $\max(\mathbf{CF})$ represent the minimum and maximum raw *CF* values in the current sequence, respectively. This normalization maps the fluctuation scores to the range $[0, 1]$, highlighting the relative instability of specific tokens compared to the rest of the generated context.

**More Statistics.** Figure 7 shows a visualization of the *CF* scores across a sample caption. Statistical analysis in Table 6 revealed that the average *CF* of hallucinated tokens exceeded the global token average in 63.74% of samples, with fine-grained tokens doing so in 58.92%. Notably, 74.83% of samples exhibited higher fluctuation for the combined set of these two types. These findings confirm *CF* as a robust measure of inherent model uncertainty, establishing a theoretical foundation for hallucination localization.

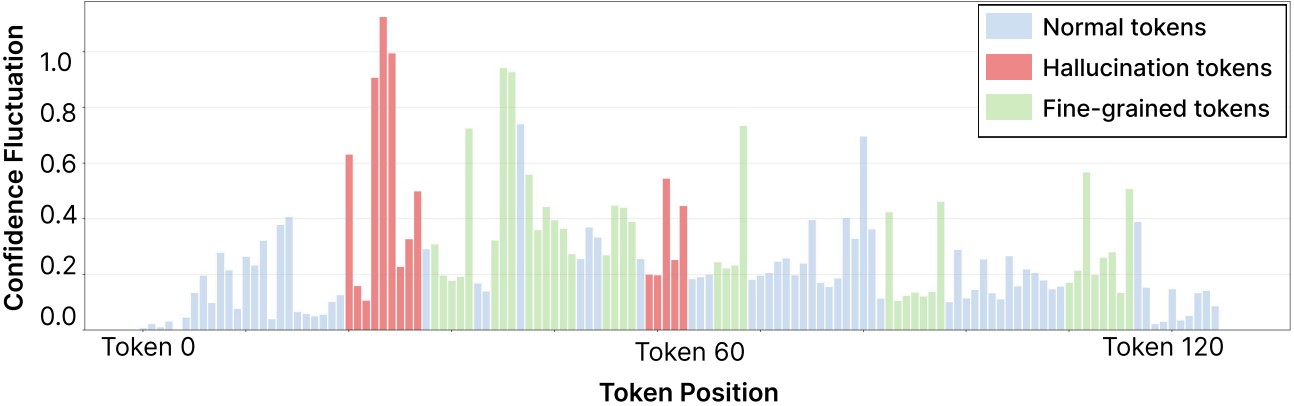

*Figure 7.* Visualization of the Confidence Fluctuation scores across a sample caption.

*Table 6.* Statistical analysis of Confidence Fluctuation (*CF*). The table shows the percentage of samples where the average *CF* of the specified token types exceeds the global token average.

| Token Type | High Fluctuation Rate (%) |
|---|---|
| Hallucinated Tokens | 63.74 |
| Fine-grained Tokens | 58.92 |
| Combined Set | 74.83 |

**Reliability of CF Across Denoising Steps.** To verify that *CF* reliably distinguishes hallucinated tokens from normal ones under different denoising steps, we evaluate the base LLaDA-V model on DetailCaps-4870 with three denoising budgets $T \in \{32, 64, 128\}$, covering both accelerated and standard inference regimes. Table 7 reports the mean normalized *CF* of three token categories (hallucinated, fine-grained, and normal), and Table 8 reports the Part-of-Speech (POS) distribution of the top-10 tokens under each metric.

The observations consistently hold across all $T$. First, hallucinated tokens exhibit higher mean normalized *CF* than normal tokens at all settings, confirming that the distributional shift observed in Figure 2 (c) is not an artifact of a particular denoising step. Second, high-*CF* tokens remain concentrated on visually meaningful entities, whereas low-confidence tokens are

*Table 7.* Mean normalized *CF* ($\mu \pm \sigma$) of hallucinated, fine-grained, and normal tokens under different denoising steps $T$ on LLaDA-V. Hallucinated tokens consistently exhibit higher *CF* than normal tokens across all settings.

| Steps ($T$) | Hallucination | Fine-grained | Normal |
|---|---|---|---|
| 128 | $0.25 \pm 0.20$ | $0.23 \pm 0.19$ | $0.20 \pm 0.18$ |
| 64 | $0.24 \pm 0.20$ | $0.22 \pm 0.19$ | $0.18 \pm 0.18$ |
| 32 | $0.21 \pm 0.20$ | $0.19 \pm 0.18$ | $0.15 \pm 0.17$ |

*Table 8.* Part-of-Speech distribution of top-10 tokens identified by High *CF* and Low Confidence under different denoising steps. High-*CF* tokens consistently concentrate on content words (Nouns, Verbs), while Low-Confidence tokens skew toward function words (Determiners, Punctuation). Det: Determiner; Adj: Adjective; Prep: Preposition; Punct: Punctuation.

| Steps ($T$) | Metric | Top 1 | Top 2 | Top 3 |
|---|---|---|---|---|
| 128 | High *CF* | Noun (53%) | Verb (13%) | Adj (9%) |
|  | Low Conf | Noun (33%) | Det (23%) | Verb (11%) |
| 64 | High *CF* | Noun (52%) | Verb (13%) | Prep (8%) |
|  | Low Conf | Noun (29%) | Det (26%) | Punct (12%) |
| 32 | High *CF* | Noun (55%) | Verb (12%) | Prep (8%) |
|  | Low Conf | Det (28%) | Noun (26%) | Punct (17%) |

dominated by syntactic function words. Together, these results indicate that *CF* captures uncertainty of vision rather than syntax, and remains applicable under accelerated sampling.

## B. Token-Level Masking Algorithm Details

To construct the VGR-Instruct dataset, we require precise alignment between the visual evidence and the textual tokens that need refinement. A simple string-based replacement may fail due to tokenization boundaries. To address this, we implement a *Token-Level Masking Algorithm*.

Algorithm 1 details the process. We first encode both the global caption and the local object description into token sequences. We then perform a sliding window search to locate the exact subsequence of token IDs corresponding to the local description. If a match is found, we replace the tokens with the special mask token; otherwise, we fall back to string matching.

## C. Hyper-parameters and Generalization

Table 9 lists the detailed configurations for the three VGR variants (LLaDA-VGR, LaViDa-VGR, and MMaDA-VGR), covering model architecture, training settings, and inference strategies. Two model-specific adaptations are worth noting. First, MMaDA does not employ a separate visual projector and instead processes image tokens directly through the language backbone; accordingly, for MMaDA-VGR we fine-tune the image-token portion of the word token embedding (`wte`) layer rather than the projector. Second, MMaDA adopts a semi-autoregressive decoding strategy, which struggles to generate coherent long sentences under step-by-step parallel decoding. To ensure the validity and fluency of the generated content, we adopt a block size of 32 and a denoising step count of 64 for MMaDA-VGR, while the other two variants follow the standard full-sequence denoising configuration.

## D. Experiment Details

**The Detailed Results on CapMAS.** As shown in Table 10, while LLaDA-VGR achieves significant gains in CLAIR (+2.85) and Coverage (+6.12), we observe a slight decline in the Factuality score (61.06 → 58.78) compared to the baseline LLaDA-V. We attribute this phenomenon to the inherent trade-off between safety and informativeness, as well as the perception limitations of the automated evaluator. The baseline LLaDA-V tends to generate generic descriptions focused on salient objects, minimizing the risk of errors but missing rich visual details. In contrast, LLaDA-VGR actively recovers fine-grained attributes and small objects, as evidenced by the substantial increase in Coverage (49.22 → 55.34). Increasing the information density naturally introduces a higher probability of minor hallucinations compared to sparse captions.

---

**Algorithm 1** Token-Level Masking Strategy

---

1: **Input:** Global Caption $T_{global}$, Local Description $T_{local}$, Tokenizer $\mathcal{T}$
2: **Output:** Masked Caption $T_{masked}$
3: *// Step 1: Tokenize inputs without special tokens*
4: $Z_{global} \leftarrow \mathcal{T}.\text{encode}(T_{global})$
5: $Z_{local} \leftarrow \mathcal{T}.\text{encode}(T_{local})$
6: $id_{mask} \leftarrow \mathcal{T}.\text{encode}(\texttt{"<|mdm\_mask|>"})$
7: **if** $\text{len}(Z_{local}) == 0$ **then**
8:     **return** $T_{global}$
9: **end if**
10: *// Step 2: Sliding window search for token sequence matching*
11: $L_g \leftarrow \text{len}(Z_{global})$
12: $L_l \leftarrow \text{len}(Z_{local})$
13: *match_found* $\leftarrow$ False
14: **for** $i = 0$ **to** $L_g - L_l$ **do**
15:     **if** $Z_{global}[i : i + L_l] == Z_{local}$ **then**
16:         *// Match found: Replace tokens with mask IDs*
17:         $Z_{masked} \leftarrow Z_{global}$
18:         $Z_{masked}[i : i + L_l] \leftarrow [id_{mask}] \times L_l$
19:         *match_found* $\leftarrow$ True
20:         **break**
21:     **end if**
22: **end for**
23: *// Step 3: Return result or fallback*
24: **if** *match_found* **then**
25:     $T_{masked} \leftarrow \mathcal{T}.\text{decode}(Z_{masked})$
26:     **return** $T_{masked}$
27: **else**
28:     *// Fallback to string-based matching if token matching fails*
29:     **return** StringMatchFallback$(T_{global}, T_{local})$
30: **end if**

---

However, the overall improvement in CLAIR indicates that the gain in semantic richness outweighs the penalty in localized precision. CapMAS relies on standard MLLMs (e.g., GPT-4o (Hurst et al., 2024)) as verifiers to check atomic propositions. Our visual-guided refinement mechanism serves as a "magnifying glass", recovering subtle details (e.g., distant background objects or complex textures) that may exceed the single-pass perception capability of the CapMAS verifier. Consequently, correct fine-grained details generated by LLaDA-VGR may be falsely flagged as hallucinations by a coarser judge. This interpretation aligns with our AMBER results, where human-annotated ground truths confirm our method's superiority in handling hard, fine-grained samples.

**Inference Efficiency.** We evaluated the inference latency on 1,004 samples from the AMBER-g benchmark, as shown in Table 11. The total inference time for LLaDA-VGR averages 38.95s per sample, compared to 18.44s for the baseline LLaDA-V. This increase stems from the iterative refinement process. Specifically, the marginal cost for each refinement round is only 3.30s. This round latency decomposes into two parts: (1) 1.75s for the auxiliary pipeline (Grounding DINO detection and syntactic anchoring), and (2) 1.55s for the MDLM generative denoising step. The high efficiency of the latter is attributed to the non-autoregressive nature of diffusion models, which enables masked regeneration: the model only needs to denoise the specific masked spans rather than regenerating the full sequence, keeping the computational cost of each correction step consistently low.

**Scalability via Batch Inference.** The VGR pipeline supports batch inference, allowing it to scale effectively for large-scale applications. We conducted a scalability test on MMHal-Bench across different batch sizes and the results are shown in Table 12.

**Construction of Difficulty Subsets on AMBER.** To provide a fine-grained analysis of the model's perception capabilities,

*Table 9.* Detailed hyper-parameters for LLaDA-VGR, LaViDa-VGR, and MMaDA-VGR. The three models share the same training and inference recipe except for the base architecture, the corresponding visual tokenizer/encoder, and the decoding configuration of MMaDA-VGR (due to its semi-autoregressive nature).

| Configuration | LLaDA-VGR | LaViDa-VGR | MMaDA-VGR |
|---|---|---|---|
| *Model Architecture* | | | |
| Base Model | LLaDA-V | LaViDa | MMaDA |
| Visual Encoder | Siglip2-so400m | Siglip-so400m | MAGVIT-v2 |
| LoRA Target Modules | Attention & FFN | Attention & FFN | Attention & FFN |
| LoRA Rank ($r$) | 64 | 64 | 64 |
| LoRA Alpha ($\alpha$) | 128 | 128 | 128 |
| LoRA Dropout | 0.05 | 0.05 | 0.05 |
| *Training Settings* | | | |
| Hardware | $4 \times$ NVIDIA A100 (80GB) | | |
| Batch Size | 4 per GPU | 4 per GPU | 4 per GPU |
| Gradient Accumulation Steps | 8 | 8 | 8 |
| Learning Rate | $1 \times 10^{-4}$ | $1 \times 10^{-4}$ | $1 \times 10^{-4}$ |
| *Inference & Refinement* | | | |
| Max Generation Length | 128 | 128 | 128 |
| Decoding Strategy | Parallel | Parallel | Semi-autoregressive |
| Block Size | 128 | 128 | 32 |
| Denoising Steps | 128 | 128 | 64 |
| Visual Gate Threshold ($\tau$) | 0.40 | 0.40 | 0.40 |
| Max Refinement Turns | 6 | 6 | 6 |
| Expansion Scale | $1.5\times$ (if area $< 0.05$) | | |

*Table 10.* The detailed scores of three metrics on CapMAS.

| Model | CapMAS | | |
|---|---|---|---|
| | CLAIR ↑ | Coverage ↑ | Factuality ↑ |
| *AR model* | | | |
| LLaVA-1.5-7B | 62.10 | 34.30 | 52.80 |
| InternVL-2.5-7B | 78.37 | 52.57 | 78.69 |
| Qwen2.5-VL-7B | 80.48 | 57.32 | 82.73 |
| *Discrete diffusion model* | | | |
| MMaDA | 35.45 | 14.33 | 57.98 |
| FUDOKI | 51.94 | 39.18 | 46.04 |
| LaViDa | 56.22 | 44.18 | 53.57 |
| LLaDA-V | 65.54 | 49.22 | **61.06** |
| LLaDA-VGR | **68.39** | **55.34** | 58.78 |

*Table 11.* Inference latency analysis.

| Model Component | Avg. Latency (s) |
|---|---|
| LLaDA-V | 18.44 |
| LLaDA-VGR (6 refinement round) | 38.95 |
|    refinement loop (per round) | 3.30 |
|    remasked token generation (per round) | 1.55 |

*Table 12.* Scalability of LLaDA-VGR under different batch sizes on MMHal-Bench. Per-sample latency decreases consistently as the batch size grows, demonstrating that the iterative refinement overhead is well-amortized in batched deployment.

| Batch Size | Latency per Sample (s) | Throughput Speedup |
|:---:|:---:|:---:|
| 1 | 24.06 | 1.00× |
| 2 | 14.52 | 1.65× |
| 4 | 10.82 | 2.22× |
| 8 | 7.67 | 3.14× |

particularly for small objects, we stratified the AMBER dataset into three difficulty levels: **Hard**, **Middle**, and **Simple**. The stratification process is based on the premise that smaller objects are intrinsically harder to ground and describe accurately. The detailed construction pipeline is as follows:

1. **Object Detection:** We utilized **Grounding DINO** to detect the bounding boxes of all ground truth objects mentioned in the annotations for each image sample.

2. **Difficulty Scoring:** For each sample containing multiple objects $\{o_1, o_2, \ldots, o_n\}$, we calculated the *Area Ratio* $(r_i)$ for each object relative to the full image. We then defined the sample-level difficulty score $\gamma$ as the **minimum** area ratio among all objects in that image:

$$\gamma = \min_i(r_i) \tag{7}$$

   This ensures that an image is classified based on its finest (most challenging) visual detail.

3. **Subset Division:** Finally, we sorted all samples in ascending order based on their difficulty scores $\gamma$ (from smallest to largest area) and divided them according to the following quantiles:

   - **Hard Subset (Top 10%):** The most challenging samples containing extremely small objects (cumulative distribution $0\% \sim 10\%$).
   - **Middle Subset (Next 40%):** Samples containing moderately sized objects (cumulative distribution $10\% \sim 50\%$).
   - **Simple Subset (Last 50%):** Samples dominated by large, prominent objects (cumulative distribution $50\% \sim 100\%$).

This rigorous sorting allows us to isolate and evaluate the model's specific improvement in fine-grained visual perception. The evaluation results are shown in Table 13.

*Table 13.* Performance comparison across difficulty levels based on object scale.

| Subset | Visual Evidence | AMBER-g | | | |
|:---:|:---:|:---:|:---:|:---:|:---:|
| | | CHAIR ↓ | Cover ↑ | Hal ↓ | Cog ↓ |
| Simple | Global only | **7.1** | **68.2** | 39.0 | 2.6 |
| | Dual-View | **7.1** | 67.7 | **38.6** | **2.5** |
| Middle | Global only | 4.9 | 62.0 | **33.1** | **1.6** |
| | Dual-View | **4.8** | **62.5** | 33.6 | 1.9 |
| Hard | Global only | 6.1 | 60.9 | **44.0** | 2.7 |
| | Dual-View | **6.0** | **61.7** | **44.0** | **2.5** |

**Remasking Window Size Analysis.** We investigate the influence of the remasking window size $k$ and the results are shown in Table 14. We observe that $k = 2$ yields the optimal performance. A window of $k = 0$ provides insufficient context for the model to stitch the correction seamlessly into the sentence, while a larger window $k = 4$ introduces unnecessary noise, leading to generation drift. A moderate expansion strikes the best balance between context awareness and modification precision.

*Table 14.* Impact of remasking window expansion ($k$).

| Window Size | CapArena CapArena-Auto | AMBER-g CHAIR ↓ | Cover ↑ | Hal ↓ | Cog ↓ |
|---|---|---|---|---|---|
| 0 | -54.83 | **6.0** | 64.4 | **37.2** | 2.3 |
| 2 | **-52.17** | **6.0** | 64.5 | **37.2** | **2.2** |
| 4 | -54.33 | 6.2 | **65.0** | 38.0 | 2.3 |

## E. Qualitative Analysis Details

In this section, we provide a detailed comparative analysis of the refinement trajectories between our High *CF* strategy and the baseline Low Confidence strategy, using the "gorilla vs. backpack" case study from the main text.

**Failure of Low Confidence.** Figure 8 illustrates the refinement process driven by the Low Confidence metric. As observed, the tokens associated with the hallucination "bags or backpacks" do not exhibit low confidence. This is because, from a language modeling perspective, "backpacks resting on rocks" is a highly plausible and common semantic collocation in the training corpus. The model is "confidently wrong"—it does not doubt its prediction despite the visual mismatch. Instead, the Low Confidence metric flags tokens such as "building" (Round 1) and "cliff" (Round 2). These tokens represent valid visual elements but may have lower probability due to synonymous alternatives (e.g., "structure" vs. "building") or syntactic transitional uncertainty. The resulting refinements act as mere paraphrasing, consuming the iteration budget without addressing the factual error.

**Success of High CF.** In contrast, as shown in Figure 6, the High *CF* metric successfully localizes the error. Although the final confidence for "backpacks" was high, the fluctuation during the denoising steps revealed the model's underlying hesitation caused by the conflict between the visual signal (dark shapes) and the language prior (static objects on rocks). This demonstrates that *CF* effectively pierces through the "confidence mask" of language priors to identify genuine visual grounding failures.

**More Cases.** To further demonstrate the versatility and robustness of LLaDA-VGR, we present additional qualitative examples in Figure 9. These cases cover a broad spectrum of visual hallucinations beyond simple object existence. For instance, in the first case (top), the model effectively corrects **attribute errors** (changing "red button" to "white button") and **identity errors** (correcting "man" to "woman"). The second case (middle) highlights the correction of complex **spatial relationships** and interactions, refining "holds the hand" to a more accurate description "rests beneath the hand". The third case (bottom) demonstrates the ability to correct **object states** and fine-grained details, such as accurately identifying "cardboard paper bags" instead of a generic "brown box". In all instances, the High *CF* metric serves as a precise locator for visual-linguistic conflicts, enabling the refinement module to generate descriptions that are faithfully grounded in the visual evidence.

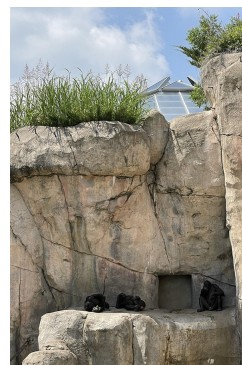

**Initial generation**
The image captures a serene outdoor scene under a clear blue sky, featuring a prominent rock formation with a large, textured cliff face. On the left, a cluster of green plants is nestled among the rocks, adding a touch of greenery to the scene. To the right, a building with a glass roof is partially visible, suggesting the presence of a structure behind the cliff. Two black objects, which appear to be bags or backpacks, are resting side by side on the rocks near the edge of the cliff. In the foreground, a few smaller rocks are visible at the base of the cliff, adding to the natural ambiance of the setting.

**Round 1**
→ Low confidence token: "building"
→ Semantic core: "a building"
→ Applied masking on: "a building with a"
→ Refinement result: "a building with a"

**Round 2**
→ Low confidence token: "cliff"
→ Semantic core: "a large, textured cliff face"
→ Applied masking on: "formation with a large, textured cliff face"
→ Refinement result: "with a large, textured cliff face"

...

**Final result**
The image captures a serene outdoor scene under a clear blue sky, featuring a prominent rock formation with a large, textured cliff face. On the left, a cluster of green plants is nestled among the rocks, adding a touch of greenery to the scene. To the right, a building with a glass roof is partially visible, suggesting the presence of a structure behind the cliff. Two black objects, which appear to be bags or backpacks, are resting side by side on a smooth, flat section of the cliff. In the foreground, a few smaller rocks are visible at the base of the cliff, adding to the natural ambiance of the setting.

*Figure 8.* Qualitative failure analysis of the Low Confidence strategy. This figure corresponds to the same sample as Figure 6 in the main text. The Low Confidence strategy fails to identify the hallucinated "bags or backpacks" because the language model assigns high probability to these tokens given the context of "resting on rocks". Instead, it targets linguistically uncertain but factually correct tokens like "building" and "cliff" for refinement. Consequently, the refinement rounds are wasted on trivial rephrasing (e.g., adding "smooth, flat section") while the core visual hallucination remains uncorrected in the final result.

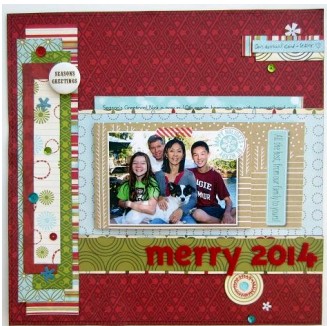

**Initial generation**
A beautifully crafted scrapbook page features a vibrant red background, adorned with a variety of decorative elements. At the top, a prominent red button reads 'Seasons greetings,' while a horizontal white strip repeats the text 'Seasons greetings.' On the left, a decorative strip includes a green section with a snowflake pattern and a blue section with a repeated 'Seasons greetings' message. Dominating the center of the page is a cheerful family photo of four people, including a man and a child with a dog. At the bottom of the page, a festive red banner displays the text 'Merry 2014.'.

**Round 1**
→ High CF token: "button"
→ Semantic core: "a prominent red button"
→ Mask on: ", a prominent red button reads"
→ Result: ", a small white button reads"

**Round 2**
→ High CF token: "child"
→ Semantic core: "a child with a dog"
→ Mask on: "man and a child with a dog."
→ Result: "child and a woman holding a dog."

...

**Final result**
A beautifully crafted scrapbook page features a vibrant red background, adorned with a variety of decorative elements. At the top, a small white button reads 'Seasons greetings,' while a horizontal white strip repeats the text 'Seasons greetings.' On the left, a decorative strip includes a green section with a snowflake pattern and a blue section with a repeated 'Seasons greetings' message. Dominating the center of the page is a cheerful family photo of four people, including a child and a woman holding a dog. At the bottom of the page, a festive red banner displays the text 'Merry 2014.'.

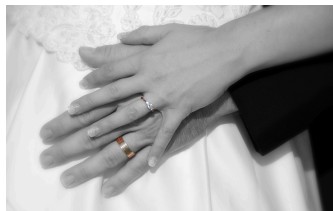

**Initial generation**
In this black and white photograph, a pair of hands, likely those of a bride and groom, are captured in a tender moment. The hand on the left, adorned with a gold ring, gently holds the hand on the right, which wears a diamond ring. These hands are resting on a white surface, possibly a sheet or a wedding dress, with a delicate lace pattern visible in the background. To the right, a black sleeve is visible, suggesting that the person with the diamond ring is wearing a dark shirt. The elbow of the black sleeve casts a soft shadow on the fabric, adding to the romantic ambiance of the scene.

**Round 1**
→ High CF token: "diamond"
→ Semantic core: "the diamond ring"
→ Mask on: "person with the diamond ring is wearing"
→ Result: "person with the gold band is wearing"

**Round 2**
→ High CF token: "holds"
→ Semantic core: "holds the hand"
→ Mask on: ", gently holds the hand on the"
→ Result: ", rests beneath the hand on the"

...

**Final result**
In this black and white photograph, a pair of hands, likely those of a bride and groom, are captured in a tender moment. The hand on the left, adorned with a gold ring, rests beneath the hand on the right, which wears a diamond ring. These hands are resting on a white surface, possibly a sheet or a wedding dress, with a delicate lace pattern visible in the background. To the right, a black sleeve is visible, suggesting that the person with the gold band is wearing a dark shirt. The elbow of the black sleeve casts a soft shadow on the fabric, adding to the romantic ambiance of the scene.

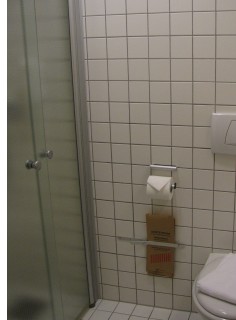

**Initial generation**
The image depicts a well-lit bathroom with a modern and clean aesthetic. On the left, a glass shower door is prominently visible, featuring a silver handle. Adjacent to the shower, a white-tiled wall forms the backdrop of the bathroom, with a toilet paper holder mounted on it. This holder holds a roll of white toilet paper, and a small brown box is visible below the toilet paper roll. Positioned against the wall, to the right of the holder, is a sleek white toilet with a visible tank and seat. The floor beneath the toilet is also covered in white tiles, contributing to the overall brightness of the space.

**Round 1**
→ High CF token: "visible"
→ Semantic core: "a visible tank"
→ Applied masking on: "toilet with a visible tank and seat"
→ Refinement result: "toilet with its lid and seat closed"

**Round 2**
→ High CF token: "box"
→ Semantic core: "a small brown box"
→ Applied masking on: "and a small brown box is visible"
→ Refinement result: "and small cardboard paper bags are visible"

...

**Final result**
The image depicts a well-lit bathroom with a modern and clean aesthetic. On the left, a glass shower door is prominently visible, featuring a silver handle. Adjacent to the shower, a white-tiled wall forms the backdrop of the bathroom, with a toilet paper holder mounted on it. This holder holds a roll of white toilet paper, and small cardboard paper bags are visible below the toilet paper roll. Positioned against the wall, to the right of the holder, is a sleek white toilet with its lid and seat closed. The floor beneath the toilet is also covered in white tiles, contributing to the overall brightness of the space.

*Figure 9.* Additional qualitative examples of LLaDA-VGR. We illustrate three diverse cases where our framework successfully corrects fine-grained hallucinations. The red text indicates the initial hallucinated content triggered by High *CF* alerts, and the blue text shows the corrected output after visual-guided refinement. These examples demonstrate the model's capability in fixing errors related to object attributes (e.g., color, material), spatial relationships, and object identification.

# F. System Prompts

In this section, we present the detailed system prompts used throughout our pipeline. Table 15 outlines the prompt for the annotation task in the empirical study, where the model identifies hallucinations and fine-grained visual details in the generated texts. Table 16 displays the instruction used for constructing the VGR-Instruct dataset, ensuring high-quality local-global correspondence. Table 17 illustrates the prompt used during the visual-guided refinement stage. In this stage, the model is tasked with restoring masked tokens by strictly adhering to the visual evidence provided in the dual-view inputs (global image and local crop), thereby correcting hallucinations.

*Table 15.* The prompt used for annotating hallucinated tokens and identifying fine-grained details. The model performs a dual task of inconsistency detection and detail extraction.

```
You are an expert editor with an extremely high attention to detail, specializing in
 image-text consistency.

## Task Overview:
You will perform TWO tasks simultaneously. Given an image and its caption:
1.  **Task A:** Find all factually *inconsistent* phrases.
2.  **Task B:** Find all factually *correct* phrases that describe *fine-grained
visual details*.
You will return a single JSON object containing two separate lists for these
findings.

## Part 1: Task A - Identifying Inconsistencies
Find every phrase in the caption that is factually *inconsistent* with the image.

**Principle: Semantic Phrase Correction**
- Your goal is to find the *smallest semantically complete phrase* that contains the
 error. This is often a noun phrase, prepositional phrase, or verb phrase.
- This is *not* limited to a single word, but should *not* be an unnecessarily long
sentence.

**Correction Rules:**
- For each inconsistency, provide the `original_text` and a `corrected_text`.
- The `corrected_text` MUST be a positive replacement (e.g., "an empty shelf").
- Do NOT use negations or corrective markers (e.g., "no", "not", "instead").
- Maintain the same grammatical role.

## Part 2: Task B - Identifying Fine-Grained Phrases
Find all phrases in the caption that are factually *correct* and describe *fine-
grained visual details*.

**Definition of "Fine-Grained":**
- These are phrases that go beyond simple object identification (e.g., not just "a
cat" or "a toilet").
- Look for descriptions of:
    - **Attributes:** Specific colors, patterns, textures (e.g., "a black and white
    checkered floor", "tabby fur", "a bright pink flamingo")
    - **Complex Relations:** Non-obvious spatial positions (e.g., "peering over the
    lid")
    - **Specific States:** (e.g., "lid is open", "stretching its front paw")

**Correction Rules:**
- These phrases MUST be factually correct.
- For these, you ONLY need to provide the `original_text`. No `corrected_text` is
needed.

## Formatting Rules:
- Return ONLY valid JSON in the following schema (no extra text).
{
  "inconsistencies": [
    {
      "original_text": "minimal inconsistent semantic phrase",
      "corrected_text": "drop-in positive replacement phrase"
    }
  ],
  "fine_grained_phrases": [
    {
      "original_text": "correct and detailed phrase"
    }
  ]
}
```

*Table 16.* The prompt used for constructing the VGR-Instruct dataset. It instructs the model to generate a coherent caption incorporating specific instance information.

```
You are an expert image captioner. Given an image and information about specific
instances in the image, generate a coherent, natural paragraph describing the image.

## Task:
Generate a complete, coherent caption for the entire image that includes
descriptions of the specified instances.

## Instance Information:
{instances_text}

## Requirements:
1. Write a single, coherent paragraph (not bullet points or separate sentences).
2. The caption should describe the entire image scene, not just the instances.
3. For each specified instance, include a natural description that is consistent
with the reference descriptions but may be rephrased for fluency.
4. The description should flow naturally and read like a human-written caption.
5. The number of instances in the output must match the number of instances provided
 ({len(instances)}).

## Output Format:
Return a JSON object with the following structure:
{
    "full_caption": "A complete textual description",
    "instances": [
        {
            "bbox": [x, y, w, h],
            "local_caption": "A partial text fragment describing this instance (it
            must be part of the complete description)"
        },
        ...
    ]
}

Important:
- The "full_caption" should be a complete, coherent paragraph describing the entire
image.
- Each "local_caption" MUST be a SHORT PHRASE (no more than 10 words, not a full
sentence) that appears in the "full_caption" and describes the corresponding
instance.
- The "local_caption" should be concise and focused on the key visual attributes of
the instance (e.g., "a tall red dog", "on the table", "bent neck").
- The "local_caption" should match the bbox position in the image.
- The order of instances in the output must match the order provided.
- Each instance's bbox should match the input bbox (values may be rounded to one
decimal place).
- Make sure the JSON is valid and properly formatted.
- Return exactly {len(instances)} instances in the instances array.
```

*Table 17.* The prompt used for the Visual-Guided Refinement stage. The model is instructed to utilize the dual-view visual evidence (global image and zoomed-in crop) to accurately restore the masked text, prioritizing visual facts over language priors.

```
You are presented with two images: the original full image and a zoomed-in visual
detail, along with a text description with missing parts.

Task: Analyze the specific visual attributes (such as texture, pattern, color, and
object shape) in the provided zoomed-in image crop, while considering the context
from the original full image.

Instruction: The masked part of the text describes this specific visual evidence
shown in the zoomed-in crop. Do not guess based on common language patterns. Instead
, look closely at the zoomed-in crop to accurately restore the missing text. What
you see in the crop is the ground truth.

The first image is the original full image, and the second image is the zoomed-in
detail.
```

