# OpenReview forum: "When Diffusion Language Models Hesitate: Detecting and Correcting Visual Hallucinations via Confidence Fluctuation"
_ICML.cc/2026/Conference — ICML 2026 regular_

### Official Review · Reviewer_8kvW · 2026-03-01

**Soundness:** 2
**Presentation:** 2
**Significance:** 3
**Originality:** 2
**Overall Recommendation:** 3
**Confidence:** 4

**Summary:**

This paper identifies that existing MDLMs suffer from severe visual hallucinations, attributed to the static nature of visual perception. To tackle this challenge, the authors propose VGR (Visual-Guided Refinement), a framework that enables MDLMs to revisit visual details by exploiting diffusion dynamics. The key insight is that the temporal trajectory of confidence during denoising reveals intrinsic uncertainty. VGR utilizes this fluctuation signal to detect uncertain spans and corrects them through targeted visual evidence extraction and in-place remasking. Extensive experiments demonstrate that the proposed method significantly reduces hallucinations.

**Compliance With Llm Reviewing Policy:**

Affirmed.

**Key Questions For Authors:**

Please see the weakness.

**Limitations:**

The authors did not discuss the limitations of their work.

**Strengths And Weaknesses:**

Strengths:

1. This paper is clearly written.

2. The authors analyzed the problem of visual hallucination in multimodal diffusion language models, pointing out that this often occurs with tokens exhibiting confidence fluctuations.

3. Comprehensive experiments and albation studies on multiple benchmarks demonstrate the effectiveness of the proposed method.

Weaknesses:

1. The paper's observation is good, but the proposed method seems more like patching a problem rather than fundamentally solving it.

2. The proposed method is inelegant and significantly increases the complexity of the system during inference, making the entire pipeline cumbersome and reducing its practical application value.

3. The proposed method requires introducing an additional Dino model and concatenating global and local visual tokens for multiple remasking and refinement processes, which increases the model's inference cost and complexity. The paper lacks a comprehensive analysis of the additional latency and memory costs of inference.

4. Can the proposed pipeline support batch inference? I think this may have potential efficiency issues for batch inference.

---

> ### Author Rebuttal · Authors · 2026-03-31
>
> Dear reviewer 8kvW,
>
> We sincerely thank you for the constructive feedback. Below are our point-by-point clarifications. Should our clarifications address your concerns, we would deeply appreciate your kind consideration in updating your score.
>
> ---
>
> **Q1&Q2: Fundamental Solution vs. "Patching"**
>
> We agree that VGR is not an architecture-level redesign of MDLMs. Its design is instead motivated by recent tool-based multimodal reasoning, where external visual tools are used to transform vision from a static input into a revisitable reasoning workspace.
>
> Our goal is to address a key structural limitation of current MDLMs: visual information is encoded only once at initialization, while the subsequent denoising process may drift toward language priors without a natural mechanism to revisit relevant image regions.
>
> Moreover, intermediate denoising states are noisy token predictions rather than stable reasoning steps, and the fixed generation format makes dynamic visual interaction difficult to incorporate directly. Under these constraints, VGR introduces an active-viewing mechanism that is tightly aligned with the diffusion model’s native remasking capability, enabling targeted visual re-grounding in a way that is compatible with iterative denoising. We will revise the manuscript to make this motivation and intended scope more explicit.
>
> ---
>
> **Q2&Q3: Complexity, Latency, and Memory Analysis**
>
> While the pipeline introduces additional components, it maintains high practical value by leveraging the non-autoregressive nature of MDLMs.
>
> - For latency analysis, we would like to clarify that we have provided detailed analysis in Table 7 in Appendix D. While the iterative process introduces a 2.1x time overhead, the marginal cost for each refinement round is only 3.30s. This is highly efficient because the model only needs to denoise specific masked spans rather than regenerating the full sequence. In addition, using batch inference can provide further speedup. We have tested on MMHal-Bench by increasing the batch size from 1 to 8, which reduces the average processing time per sample from 24.06s to 7.67s.
> - Regarding memory overhead, we conducted a peak memory analysis. At Batch Size = 1, the peak memory usage of the LLaDA-VGR pipeline is 28.29 GB, compared to 23.50 GB for the base LLaDA-V model during full-sequence generation. Although external tools introduce a modest increase in memory consumption, this overhead is well-justified by the substantial gains in visual perception accuracy. Importantly, VGR does not introduce a severe memory bottleneck and remains fully compatible with standard hardware configurations.
>
> ---
>
> **Q4: Batch Inference Support and Efficiency**
>
> The VGR pipeline supports batch inference, allowing it to scale effectively for large-scale applications. We conducted a scalability test on MMHal-Bench across different batch sizes and the results are shown below. We will include these test in the final version.
>
> | **Batch Size** | **Latency per Sample (s)** | **Throughput Speedup** |
> | -------------- | -------------------------- | ---------------------- |
> | **1**          | 24.06                      | 1.0x                   |
> | **2**          | 14.52                      | 1.65x                  |
> | **4**          | 10.82                      | 2.22x                  |
> | **8**          | 7.67                       | 3.14x                  |
>
> ---
>
> **Discussion of Limitations**
>
> We sincerely thank the reviewer for the reminder regarding the discussion of limitations. Due to the strict page limits of the initial submission, we were unable to include a dedicated section for this. We commit to incorporating a comprehensive "Limitations" section in the final version, where we will objectively evaluate our work from multi-perspectives, such as dependency on external tools and data requirements.
>
> ---
>
> Thank you again for your insightful comments and the time devoted to our work.
>
> Best regards,
>
> Authors

---

> > ### Author Rebuttal · Reviewer_8kvW · 2026-04-03
> >
> > The proposed method is not elegant. It significantly increases the complexity of the inference pipeline, raises the difficulty of practical deployment, and reduces its application value. Introducing an additional model incurs substantially higher inference costs, including both latency and memory consumption.

---

> > > ### Author Response · Authors · 2026-04-03
> > >
> > > Dear Reviewer 8kvW,
> > >
> > > We sincerely thank you for the continued engagement. We would like to address the concern about elegance and deployment complexity directly.
> > >
> > > ---
> > >
> > > **VGR leverages MDLM-intrinsic mechanisms, not external patches.** The perceived complexity stems from a fundamental structural constraint of MDLMs: static visual encoding at initialization prevents the model from re-examining image regions during denoising. An effective solution to this limitation therefore requires introducing visual re-interaction at inference time, or substantial architectural changes to the base model. VGR's design philosophy is to accomplish this with the minimum necessary intervention: (1) CF is derived entirely from the diffusion model's own denoising trajectory, requiring no auxiliary model for uncertainty localization; (2) Grounding DINO bridges the MDLM's inability to generate spatial coordinates and is lightweight (peak: 999.48 MB); (3) the correction itself exploits the MDLM's **native remasking capability**, requiring no architectural modification. The added modules are thus not arbitrary complexity, but targeted solutions to well-defined gaps imposed by the MDLM paradigm.
> > >
> > > **Inference-time computation for perceptual fidelity is a recognized standard in the field.** VGR is situated within the "Thinking with Images" paradigm [1], an established and rapidly growing research direction that explicitly recognizes inference-time visual re-interaction as the necessary mechanism for overcoming static perception limits. This paradigm is well-represented at top venues: works such as VPT [2] (ICCV 2025), AGILE [3] (ICLR 2026), and PaDT [4] (ICLR 2026) all introduce external tools or additional modules at inference time as a deliberate design choice, analogous to how Chain-of-Thought reasoning accepts higher token cost to break reasoning ceilings. The survey [1] explicitly characterizes such overhead as an accepted trade-off for perceptual fidelity. VGR's overhead is within this established standard, and is further amortized by batch inference: at batch size 8, per-sample latency reduces to **7.67s**, making practical deployment feasible.
> > >
> > > **MDLM research is at an early stage where reliability precedes architectural purity.** We view VGR as a concrete step toward principled active visual reasoning in MDLMs. Streamlining the pipeline, for instance by distilling the refinement capability into the base model, is a natural next step that we will explicitly discuss in the final version.
> > >
> > > [1] Su Z, Xia P, Guo H, et al. Thinking with images for multimodal reasoning: Foundations, methods, and future frontiers[J]. arXiv preprint arXiv:2506.23918, 2025.
> > >
> > > [2] Yu R, Ma X, Wang X. Introducing visual perception token into multimodal large language model[J]. arXiv preprint arXiv:2502.17425, 2025. (ICCV 2025)
> > >
> > > [3] Zeng Y, Huang W, Huang S, et al. Agentic Jigsaw Interaction Learning for Enhancing Visual Perception and Reasoning in Vision-Language Models[J]. arXiv preprint arXiv:2510.01304, 2025. (ICLR 2026)
> > >
> > > [4] Su Y, Zhang H, Li S, et al. Patch-as-decodable-token: Towards unified multi-modal vision tasks in mllms[J]. arXiv preprint arXiv:2510.01954, 2025. (ICLR 2026)
> > >
> > > ---
> > >
> > > Thank you again for your insightful comments and the time devoted to our work.
> > >
> > > Best regards,
> > >
> > > Authors

---

### Official Review · Reviewer_YsfF · 2026-03-09

**Soundness:** 3
**Presentation:** 4
**Significance:** 3
**Originality:** 3
**Overall Recommendation:** 4
**Confidence:** 3

**Summary:**

See Strengths&Weakness

**Compliance With Llm Reviewing Policy:**

Affirmed.

**Key Questions For Authors:**

1. From  Figure 2 , the CF fluctuations of hallucinated tokens do not appear to be significantly stronger compared to those of  normal tokens. Could the authors clarify this observation?
2. I am concerned about the  efficiency of the proposed method , including both  training and inference costs .
3. The authors constructed a 37k dataset for fine-tuning to improve local detection capability. Could this approach potentially  reduce the model's ability to detect hallucinations at the global level ?

**Limitations:**

1. The method  requires constructing a high-quality local–global dataset in advance , which may limit its practicality.

**Strengths And Weaknesses:**

Strengths

1. The paper proposes a Confidence Fluctuation (CF)-based framework to address hallucination, which is different from traditional confidence-based methods and demonstrates a certain degree of novelty.
2. The observation regarding the differences in CF patterns between normal tokens and hallucinated tokens is very interesting.
3. The paper is  clearly written and well structured .

Weaknesses

1. Although the proposed method can effectively mitigate hallucinations, it  requires constructing a high-quality local–global dataset in advance , which may limit its practicality.

---

> ### Author Rebuttal · Authors · 2026-03-31
>
> Dear Reviewer YsfF,
>
> We sincerely thank you for the insightful comments. To address the concerns proposed, we have provided detailed clarifications below.
>
> ---
>
> **Q1: Statistical Significance of CF (Figure 2)**
>
> We thank the reviewer for this careful observation, and would like to offer a more complete reading of Figure 2(c). First, Figure 2(c) is a probability density plot, which reflects the overall distributional tendency of each token type after normalization rather than absolute counts. Accordingly, the evidence is better interpreted from the distributional trend as a whole, rather than from a large visual gap at every point along the curve. In particular, normal tokens are much more concentrated in the near-zero CF region, indicating that many of them remain highly stable throughout denoising, whereas hallucinated tokens are less concentrated in this highly stable region and more likely to shift toward relatively higher CF values. Second, to further verify that this is not an accidental visual effect, we additionally evaluated the base LLaDA-V under different denoising schedules $T \in \\{32, 64, 128\\}$. Across all settings, hallucinated tokens consistently show higher mean normalized CF than normal tokens. We will revise the manuscript to clarify that CF is intended to prioritize visually unstable positions for further verification and refinement.
>
> | **Steps (T)** | **Hallucination (μ±σ)** | **Fine-grained (μ±σ)** | **Normal (μ±σ)** |
> | ------------- | ----------------------- | ---------------------- | ---------------- |
> | **128**       | 0.25 $\pm$ 0.20         | 0.23 $\pm$ 0.19        | 0.20 $\pm$ 0.18  |
> | **64**        | 0.24 $\pm$ 0.20         | 0.22 $\pm$ 0.19        | 0.18 $\pm$ 0.18  |
> | **32**        | 0.21 $\pm$ 0.20         | 0.19 $\pm$ 0.18        | 0.15 $\pm$ 0.17  |
>
> ---
>
> **Q2: Efficiency and Costs**
>
> - **Inference Efficiency:** The reported 2.1x latency overhead corresponds to the full VGR pipeline with 6 refinement rounds. Each round only performs localized refinement, costing about 3.30s, of which 1.55s is spent on remasked token generation. In addition, the pipeline supports batch inference and shows good throughput scalability in practice: on MMHal-Bench, increasing the batch size from 1 to 8 reduces the average processing time per sample from 24.06s to 7.67s.
> - **Training Costs:** The VGR-Instruct dataset (37k) is relatively modest in scale comparing with the existing hallucination correction or fine-grained visual concept perception dataset, such as ViCrit (875k) [1] and ShareGPT4V (1.2M) [2]. Training was completed taking about 12 hours for 3 epochs on a single node with 4x A100 GPUs. This indicates that the additional training cost is moderate rather than prohibitive.
>
> [1] Wang X, Yang Z, Feng C, et al. ViCrit: A Verifiable Reinforcement Learning Proxy Task for Visual Perception in VLMs[C]//The Thirty-ninth Annual Conference on Neural Information Processing Systems.
>
> [2] Chen L, Li J, Dong X, et al. Sharegpt4v: Improving large multi-modal models with better captions[C]//European Conference on Computer Vision. Cham: Springer Nature Switzerland, 2024: 370-387.
>
> ---
>
> **Q3: Impact on Global Hallucination Detection**
>
> We specifically considered this issue when designing the training strategy. Rather than training only on local refinement examples, we adopt a mixed objective that combines dual-view refinement data with global caption generation to preserve broad-scene understanding. As shown in Figure 5, Dual-View remains on par with Global-Only on the Simple subset while providing clearer gains on the Hard subset.
>
> Our ablation results also suggest no degradation in global grounding capability. In Table 3, even the Global-Only strategy already improves over the base model, and the Dual-View strategy further improves local-detail perception while maintaining nearly identical performance on the global metrics:
>
> | **Strategy**    | **CapArena**   | **AMBER-g** |        |      |      |
> | --------------- | -------------- | ----------- | ------ | ---- | ---- |
> |                 | CapArena-Auto↑ | CHAIR↓      | Cover↑ | Hal↓ | Cog↓ |
> | **Base**        | -77.17         | 8.2         | 61.8   | 44.9 | 4.2  |
> | **Global only** | -53.67         | 6.1         | 64.4   | 37.2 | 2.2  |
> | **Dual-View**   | -52.17         | 6.0         | 64.5   | 37.2 | 2.2  |
>
> We will include additional reporting and discussion on efficiency and cost in the final version.
>
> ---
>
> **W1 & Limitation: Practicality of constructing the local–global dataset**
>
> While the method requires a local-global dataset, we have ensured high practicality through automation. The 37k samples were constructed using an automated pipeline leveraging existing annotations (RefCOCO+) and advanced MLLMs (Gemini). We will open-source the VGR-Instruct dataset and the construction scripts to support future research.
>
> ---
>
> Thank you again for your insightful comments and the time devoted to our work.
>
> Best regards,
>
> Author

---

> > ### Author Rebuttal · Reviewer_YsfF · 2026-04-07
> >
> > keep rating.

---

> > > ### Author Response · Authors · 2026-04-07
> > >
> > > Dear Reviewer YsfF,
> > >
> > > Thank you for your feedback and for confirming that your concerns have been fully resolved. We sincerely appreciate the time you dedicated to reviewing our rebuttal and additional experiments. Your constructive comments have been very helpful in improving the quality of our work.
> > >
> > > Best regards,
> > >
> > > Authors

---

### Official Review · Reviewer_PNvY · 2026-03-09

**Soundness:** 3
**Presentation:** 3
**Significance:** 2
**Originality:** 2
**Overall Recommendation:** 5
**Confidence:** 4

**Summary:**

This paper addresses the critical issue of visual hallucinations in Multi-modal Diffusion Language Models (MDLMs), which often arise because the denoising process drifts toward language priors due to fixed, static visual features. The authors propose VGR (Visual-Guided Refinement), a framework that leverages the temporal dynamics of the diffusion process. The core innovation is the Confidence Fluctuation (CF) metric, which identifies "hesitation" in the denoising trajectory—where correctly grounded tokens converge smoothly while hallucinated ones oscillate significantly. VGR uses CF to localize uncertain spans, retrieves local visual evidence via an external detector (Grounding DINO), and performs in-place re-masking for correction.

**Compliance With Llm Reviewing Policy:**

Affirmed.

**Final Justification:**

Thanks for the detailed rebuttal and experiments. My concerns have been adequately addressed, and i will raise my score to 5.

**Key Questions For Authors:**

1. How sensitive is the $CF$ metric to the number of total denoising steps? Would it remain reliable in accelerated sampling scenarios (e.g., < 50 steps)?
2. Regarding the Visual Gate, how was the threshold $\tau = 0.40$ determined, and what is the impact of varying this threshold on the "over-correction" rate?
3. Can the dual-view perception training generalize to other MDLMs without requiring specific fine-tuning on VGR-Instruct for each base model?
Others see weaknesses.

**Limitations:**

Yes

**Strengths And Weaknesses:**

Strengths:
1. Novel Uncertainty Signal: The insight that the temporal trajectory of confidence is more informative than the final state is highly relevant to diffusion models.
2. Architectural Simplicity: The framework requires no architectural modifications to the base MDLM, utilizing the model’s native re-masking capabilities.
3. Effective Fine-tuning: The creation of the VGR-Instruct dataset (37k samples) empowers the model with "dual-view" perception (global context + local details).
4. Superior Performance: Achieving SOTA among MDLMs on benchmarks like AMBER and MMHal-Bench, effectively closing the gap with autoregressive (AR) models.

Weaknesses:
1. The iterative refinement process introduces a 2.1x time overhead compared to the baseline.
2. The framework’s precision is inherently bottlenecked by the accuracy of Grounding DINO and the spaCy parser. If the detector fails to localize objects in complex backgrounds, the error correction logic will face challenges (despite the visual gating mechanism, it remains a passive defense).
3. Results on CapMAS show a slight decline in factuality scores as the model attempts to recall more (and riskier) fine-grained details.

---

> ### Author Rebuttal · Authors · 2026-03-31
>
> Dear Reviewer PNvY,
>
> We sincerely thank you for the constructive feedback and have added relevant experiments to address the concerns raised.
>
> ---
>
> **Q1: CF sensitivity to the number of denoising steps**
>
> To evaluate the reliability of the CF metric across different numbers of total denoising steps, we tested the base LLaDA-V model with $T \in \\{32, 64, 128\\}$ steps. As shown below, hallucinated tokens consistently yield higher normalized CF than normal tokens across all settings, including steps < 50. This indicates that CF remains reliable in accelerated sampling scenarios.
>
> | **Steps (T)** | **Hallucination (μ±σ)** | **Fine-grained (μ±σ)** | **Normal (μ±σ)** |
> | ------------- | ----------------------- | ---------------------- | ---------------- |
> | **128**       | 0.25 $\pm$ 0.20         | 0.23 $\pm$ 0.19        | 0.20 $\pm$ 0.18  |
> | **64**        | 0.24 $\pm$ 0.20         | 0.22 $\pm$ 0.19        | 0.18 $\pm$ 0.18  |
> | **32**        | 0.21 $\pm$ 0.20         | 0.19 $\pm$ 0.18        | 0.15 $\pm$ 0.17  |
>
> Part-of-Speech (POS) analysis further supports the robustness of CF. As shown in the table, high-CF signals remain concentrated on visually meaningful entities (Nouns/Verbs), whereas low-confidence tokens target syntactic noise (Determiners/Punctuation). This further supports our claim that CF captures visual epistemic uncertainty rather than generic linguistic uncertainty. We will incorporate these step-sensitivity analyses into the final version.
>
> | **Steps (T)** | **Metric** | **Top 1**  | **Top 2**  | **Top 3**   |
> | ------------- | ---------- | ---------- | ---------- | ----------- |
> | **128**       | High CF    | Noun (53%) | Verb (13%) | Adj (9%)    |
> |               | Low Conf   | Noun (33%) | Det (23%)  | Verb (11%)  |
> | **64**        | High CF    | Noun (52%) | Verb (13%) | Prep (8%)   |
> |               | Low Conf   | Noun (29%) | Det (26%)  | Punct (12%) |
> | **32**        | High CF    | Noun (55%) | Verb (12%) | Prep (8%)   |
> |               | Low Conf   | Det (28%)  | Noun (26%) | Punct (17%) |
>
> ---
>
> **Q2&W2: Visual gate threshold and detector bottleneck**
>
> We selected $\tau = 0.4$ as a trade-off between correction recall and robustness to detector errors. To evaluate this, we tested thresholds on AMBER using three metrics: Cover (measures object coverage ); Correction Rate ($\frac{rectified \ hallucinations}{total \ hallucinations}$); and Over-correction Rate ($\frac{mistakenly \ modified \ correct \ tokens}{total \ tokens}$).
>
> | **Threshold (τ)** | **Cover** | **Correction Rate** | **Over-correction Rate** |
> | ----------------- | --------- | ------------------- | ------------------------ |
> | **0.2**           | 65.2      | 97.2%               | 0.536%                   |
> | **0.4**           | 65.1      | 97.4%               | 0.521%                   |
> | **0.6**           | 64.9      | 93.3%               | 0.532%                   |
>
> As shown, lower thresholds increase over-correction risk due to detector noise. Conversely, higher thresholds cause the model to skip subtle details, reducing the correction rate. The value of 0.4 provides the balance.
>
> Regarding the dependence on Grounding DINO and spaCy, we agree this is a limitation. However, the pipeline is not purely detector-driven: if local evidence is unreliable, the refinement will proceed with the global image context. In addition, proportional crop expansion and iterative refinement reduce error propagation from a single imperfect localization step.
>
> ---
>
> **Q3: Generalization to Other MDLMs**
>
> Thank you for the insightful suggestion on the model generalization. Our method is naturally compatible with discrete masked diffusion models. To demonstrate its generalizability, we have migrated both the dual-view perception training and the VGR inference framework to LaViDa and MMaDA. While the implementation pipeline is fully operational and models are currently undergoing training, final results were not available in time for this rebuttal. We commit to incorporating these results in the final version to objectively validate the broad applicability of VGR.
>
> ---
>
> **W1 & W3: Efficiency and Factuality Trade-offs**
>
> - **Efficiency (W1):** Although the iterative process introduces a 2.1x overhead , the non-autoregressive nature of MDLMs allows for masked regeneration. The marginal cost for each refinement round is only 3.30s, keeping the computational burden significantly lower than full-sequence regeneration.
> - **Factuality (W3):** The slight decline in Factuality is a recall-precision trade-off. While the base model avoids errors by being vague, VGR recovers fine-grained attributes. The overall improvement in CLAIR indicates that the increase in semantic richness outweighs the minor penalty in factuality.
>
> ---
>
> Thank you again for your insightful comments and the time devoted to our work.
>
> Best regards,
>
> Authors

---

> > ### Author Rebuttal · Reviewer_PNvY · 2026-04-05
> >
> > Thanks for the detailed rebuttal and experiments. My concerns have been adequately addressed, and I will raise my score to 5.

---

> > > ### Author Response · Authors · 2026-04-06
> > >
> > > Dear Reviewer PNvY,
> > >
> > > Thank you very much for your positive feedback and for acknowledging our efforts in the rebuttal. We are glad to hear that our responses and additional experiments have adequately addressed your concerns.
> > >
> > > We truly appreciate your time and the constructive suggestions, which have helped improve the quality of our work.
> > >
> > > Best regards,
> > >
> > > Authors

---

### Decision · Program_Chairs · 2026-04-30

**Decision:**

Accept (regular)

**Comment:**

This paper studies hallucination behavior in multi-modal diffusion language models (MDLMs) and proposes VGR, a framework that detects and corrects hallucinated tokens by leveraging confidence fluctuation during the denoising process. The core idea of using temporal confidence dynamics as an uncertainty signal is interesting and leads to a practical method for mitigating hallucinations.

The paper received three reviews, with two generally positive assessments and one negative. After the rebuttal and discussion, one reviewer increased their score to 5, while the negative reviewer maintained their stance. All reviewers participated in the discussion, and the authors provided additional experiments and clarifications, which were taken into account by the AC.

Overall, the reviewers’ concerns mainly focus on the following aspects: (1) the complexity and efficiency of the inference pipeline, including the use of iterative refinement and external modules; (2) the dependency on auxiliary components (e.g., object detectors and parsing tools), which may affect robustness and practicality; and (3) the extent to which the method provides a fundamental solution versus a system-level refinement.

The reviewer with a negative stance particularly emphasized that the increased complexity of the inference pipeline and the perceived lack of elegance are notable drawbacks. The AC agrees that inference complexity is an important consideration. However, it is also worth noting that recent trends in the field increasingly adopt agent-style or tool-augmented pipelines to enhance model capabilities across complex tasks. In this context, a certain level of additional complexity can be considered acceptable, especially when it leads to substantial gains in performance and new insights into model behavior.

As for the concern regarding the “elegance” of the method, this is subjective and difficult to assess. From the AC’s perspective, the proposed framework is reasonably presented, technically sound, and grounded with empirical observation. Moreover, the work provides an interesting perspective on diffusion-based multimodal models and is likely to stimulate further discussion and research in this direction.

Based on the overall balance of strengths and weaknesses, and considering the positive evaluations and the authors’ effective rebuttal, the AC recommends acceptance.